# Resolving Oversmoothing with Opinion Dissensus

## Abstract

While graph neural networks (GNNs) have allowed researchers to successfully apply neural networks to non-Euclidean domains, deep GNNs often exhibit lower predictive performance than their shallow counterparts. This phenomena has been attributed in part to oversmoothing, the tendency of node representations to become increasingly similar with network depth. In this paper we introduce an analogy between oversmoothing in GNNs and consensus (i.e., perfect agreement) in multi-agent systems literature. We show that the message passing algorithms of several GNN models are equivalent to linear opinion dynamics in multi-agent systems, which have been shown to converge to consensus for all inputs regardless of the initial state. This new perspective on oversmoothing motivates the use of nonlinear opinion dynamics as an inductive bias in GNN models. In addition to being more general than the linear opinion dynamics model, nonlinear opinion dynamics models can be designed to converge to dissensus for general inputs. Through extensive experiments we show that our Behavior-inspired message passing (BIMP) neural network resists oversmoothing beyond 100 time steps and consistently outperforms existing continuous time GNNs even when amended with oversmoothing mitigation techniques. We also show several desirable properties including well behaved gradients and adaptability to homophilic and heterophilic datasets.

## 1 Introduction

A broad class of real-world systems can be naturally represented using graphs. In molecules, atoms can be represented by nodes and atomic bonds can be represented by edges (Fang et al., 2022); in animal groups, individuals can be represented by nodes and their proximity can be represented by edges (Young et al., 2013); and in transportation networks, bus stops can be represented by nodes and public transit routes can be represented by edges (Madamori et al., 2021). Because of their broad applicability, the classification, regression, and generation of graphs are of strong interest across scientific communities. While MLPs can be adapted to operate on graph data, graph neural networks (GNNs) are specifically designed to respect the graph permutation symmetry (i.e., equivariance to node relabeling) and can therefore learn generalizable graph representations, where MLPs can not.

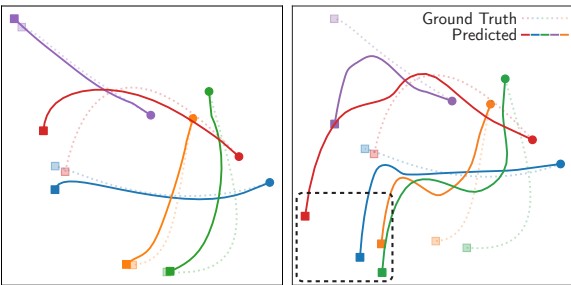

Figure 1: **Oversmoothing in multi-agent trajectory prediction.** (**Left**) Our method predicts well-behaved trajectories without oversmoothing. (**Right**) GCN-based GraphODE exhibits oversmoothing, with trajectories converging to similar solutions at longer horizons (highlighted). ●: initial state, ■: final state.

GNNs have been widely applied to fields such as recommendation (Ying et al., 2018), molecular prediction (Wang et al., 2022b), protein design (Jha et al., 2022), and complex physical system modeling (Pfaff et al., 2020). However, since node representations in GNNs can become increasingly similar with network depth; a phenomena known as oversmoothing (Li et al., 2018; Oono & Suzuki, 2019; Nt & Maehara, 2019), their performance deteriorates with increased network depth. This phenomena is illustrated in Figure 1 where we compare the recurrent application of our model

and GraphODE (Poli et al., 2019) for trajectory prediction. As a consequence of oversmoothing, GraphODE predictions tend toward the same spatial region over time (additional details are provided in Appendix E.1), whereas our predictions do not.

A number of approaches have been proposed to address the issue of oversmoothing, including the use of residual connections (Li et al., 2018; Chen et al., 2020; Liu et al., 2020; Fu et al., 2022), feature normalization (Zhao & Akoglu, 2019; Zhou et al., 2020; 2021), and alternative architectures (Chiang et al., 2019; Abu-El-Haija et al., 2019; Zeng et al., 2019). While many alternative architectures work to incorporate higher-order features (Chien et al., 2020; Chamberlain et al., 2021b; Liu et al., 2024; Li et al., 2024), continuous-depth GNNs instead interpret conventional GNN architectures as discretizations of a continuous process (Poli et al., 2019; Chamberlain et al., 2021a; Eliasof et al., 2021). This interpretation allows for the integration of techniques developed for modeling and analyzing dynamical systems (Thompson & Stewart, 2002; Brunton et al., 2016; Paredes et al., 2024; Richards et al., 2024). For example, GraphCON (Rusch et al., 2022) avoids oversmoothing by enforcing stability conditions on hidden states of coupled and damped oscillator systems; and ACMP (Wang et al., 2022a) avoids oversmoothing by introducing repulsive forces traditionally used to control clustering in particle systems. These are just a few that have shown physical inductive biases can mitigate oversmoothing while maintaining expressivity (Han et al., 2023).

In this paper, we propose a continuous-depth GNN inspired by behavioral interaction in multi-agent systems (MAS), instead of physical processes. First, we introduce an analogy likening node features in a GNN to opinions in an opinion dynamics model, feature aggregation to opinion exchange, and graph outputs to opinion outcomes (Section 3). Using this analogy, we show that oversmoothing will occur in all GNN models with layer-wise aggregation schemes that are equivalent to linear opinion dynamics. (Section 4). With this new understanding of oversmoothing, we leverage the nonlinear opinion dynamics model introduced in Leonard et al. (2024); Bizyaeva et al. (2022) to design a novel continuous-depth GNN that is provably robust to oversmoothing. In addition, we show our GNN has desirable characteristics such as well behaved gradients and adaptability to heterophilic datasets (Section 5). Finally, we empirically validate our Behavior-Inspired Message Passing (BIMP) GNN on several datasets and against competitive baselines (Section 6).

## 2 RELATED WORK

**Oversmoothing in GNNs.** Contrary to conventional feed forward networks (Montufar et al., 2014; LeCun et al., 2015), deep discrete GNNs suffer performance degradation from oversmoothing of node features (Li et al., 2018; Oono & Suzuki, 2019; Nt & Maehara, 2019). A number of analyses have been proposed to understand the oversmoothing phenomena. In linear GNNs, the addition of network layers has been shown to increase denoising and mixing effects which lead to oversmoothing (Wu et al., 2022), the residual connectivity with careful weights initialization can prevent total collapse (Scholkemper et al., 2024); GCNs (Kipf & Welling, 2016) learns representations that attempts to counteract an inherently oversmoothing prone network structure (Yang et al., 2020); and in attention based networks like GAT (Veličković et al., 2017) oversmoothing has been show to occur at an exponential rate due to the ergodicity of infinite matrix products (Wu et al., 2023). Other works, have characterized oversmoothing by energy minimization of gradient flows (Di Giovanni et al., 2022), representational rank collapse (Roth et al., 2024), theoretical bound on the convergence of energy in term on the Laplacian, weights, and activation functions (Cai & Wang, 2020), and exceeding a theoretical limit of smoothing in mean aggregation (Keriven, 2022).

**Continuous-depth GNNs.** Continuous-depth networks such as NeuralODE (Chen et al., 2018) define the network depth implicitly through the simulation of differential equations. GDE (Poli et al., 2019) leverages this notion to construct continuous-depth GNNs by propagating inputs through continuum of GNN layers governed by an underlying ODE. In order to better control and understand node dynamics, several works focuse on leveraging physical inductive biases such as heat diffusion (Chamberlain et al., 2021a), Beltrami flows (Chamberlain et al., 2021b), wave equations (Eliasof et al., 2021), coupled damped oscillators (Rusch et al., 2022), energy source terms (Thorpe et al., 2022), Allen-Cahn reaction diffusion processes (Wang et al., 2022a), blurring-sharpening forces (Choi et al., 2023), oscillator synchronization (Nguyen et al., 2024), and Ricci flow (Chen et al., 2025). These dynamics provide a principled approach to counteract known limitations of discrete GNNs including oversmoothing in deep networks and poor performance on heterophilic graphs (Han et al., 2023).

**Opinion dynamics in multi-agent systems.** Opinions can be interpreted as the preferences of agents in a multi-agent system, and provide a means by which to interpret and predict agent behavior. Specifically, consensus dynamics models (Bullo, 2018; Becchetti et al., 2020) are commonly used in multi-agent settings such as coordinating multi-vehicle movements (Justh & Krishnaprasad, 2005; Leonard et al., 2010), understanding network systems (Leonard et al., 2007; Ballerini et al., 2008), and learning on graphs (Zhou et al., 2024). Becchetti et al. (2020) provided an overview of discrete-time consensus methods and analyzed their convergence time to consensus, computational capabilities, and robustness to malicious information. However, linear models of opinion formulation can only model settings where agent opinions converge to consensus (Altafini, 2012; Dandekar et al., 2013). This short coming is resolved through the use of nonlinear opinion dynamics (Leonard et al., 2024). The nonlinearity in this model results in bifurcations Golubitsky et al. (2012) allowing opinions to evolve to dissensus rapidly even under weak input signals (Bizyaeva et al., 2022). Nonlinear opinion dynamics have been shown to able to model systems such as group decision making (Leonard et al., 2021; Bizyaeva et al., 2024; Arango et al., 2024); multi-agent control (Leonard et al., 2010; Montes de Oca et al., 2010); and relational inference (Yang et al., 2024).

## 3 GRAPH NEURAL NETWORKS AND OPINION DYNAMICS

In this section, we introduce our GNN-opinion dynamics (GNN-OD) analogy. We begin with a review GNNs and opinion dynamics, followed by a brief discussion of bifurcation theory in the context of the nonlinear opinion dynamics model (Leonard et al., 2024). We then develop an analogy likening GNNs to opinion dynamics models, and oversmoothing to opinion consensus.

### 3.1 GRAPH NEURAL NETWORKS AND OVERSMOOTHING

**Graph neural networks.** Let $\mathcal{G} = (\mathcal{V}, \mathcal{E})$ be a graph with $n = |\mathcal{V}|$ nodes and $m = |\mathcal{E}|$ edges, where an edge $e_{ij}$ exists in $\mathcal{E}$ if the nodes $\mathbf{x}_i$ and $\mathbf{x}_j$ are connected in $\mathcal{G}$. Given an input graph, a graph neural network (GNN) $f : \mathcal{G} \to \mathcal{Y}$ returns a label (or label set) over edges, nodes, or the entire graph. Of the existing GNN algorithms, a large subset can be described in the message passing framework (Gilmer et al., 2017). In this framework, layer-wise transformations are determined by learned message and update functions. The message function at layer $l$, $M^l$, and update function at layer $l$, $U^l$, are of the form,

$$\mathbf{m}_i^{l+1} = \sum_{j \in \mathcal{N}(i)} M^l(\mathbf{x}_i^l, \mathbf{x}_j^l, \mathbf{e}_{ij}), \quad \text{and} \quad \mathbf{x}_i^{l+1} = U^l(\mathbf{x}_i^l, \mathbf{m}_i^{l+1}), \tag{1}$$

where $\mathbf{x}_i^l$ denotes the representation of node $i$ at layer $l$.

**Oversmoothing.** In the GNN literature, oversmoothing is defined as the tendency for node features to become increasingly similar with increasing network depth (Rusch et al., 2023). This phenomena has been observed in discrete (Wu et al., 2022; Yang et al., 2020; Wu et al., 2023; Keriven, 2022) and continuous-depth (Chamberlain et al., 2021a; Eliasof et al., 2021; Xhonneux et al., 2020) GNNs, and correlates with reduced predictive performance. We can measure the degree of oversmoothing using the Dirichlet energy (Rusch et al., 2023; Cai & Wang, 2020), which is defined at layer $l$ as

$$E(\mathbf{X}^l) = \frac{1}{n} \sum_{\mathbf{x}_i \in \mathcal{V}} \sum_{\mathbf{x}_j \in \mathcal{N}(\mathbf{x}_i)} \|\mathbf{x}_i^l - \mathbf{x}_j^l\|_2^2, \tag{2}$$

where $\mathbf{X}^l = [\mathbf{x}_1^l, \cdots, \mathbf{x}_n^l]^T$. If the Dirichlet energy tends to zero as $l$ tends to infinity, that is,

$$\lim_{l \to \infty} \|\mathbf{x}_i^l - \mathbf{x}_j^l\|_2^2 = 0 \text{ for all } e_{ij} \in \mathcal{E}, \tag{3}$$

the network is said to exhibit oversmoothing.

### 3.2 OPINION DYNAMICS AND OPINION CONSENSUS

Let $\mathcal{M} = (\mathcal{G}^a, \mathcal{G}^o)$ be a multi-agent system with a communication graph $\mathcal{G}^a = (\mathcal{V}^a, \mathcal{E}^a, \mathbf{A}^a)$, and an option graph $\mathcal{G}^o = (\mathcal{V}^o, \mathcal{E}^o, \mathbf{A}^o)$. In this system, each of the $N_a$ agents has a real-valued opinion on each of the $N_o$ options. The adjacency matrix of the communication graph, $\mathbf{A}^a = [a_{ik}^a] \in \mathbb{R}^{N_a \times N_a}$, defines the communication strength between agents, and the adjacency matrix of the option graph,

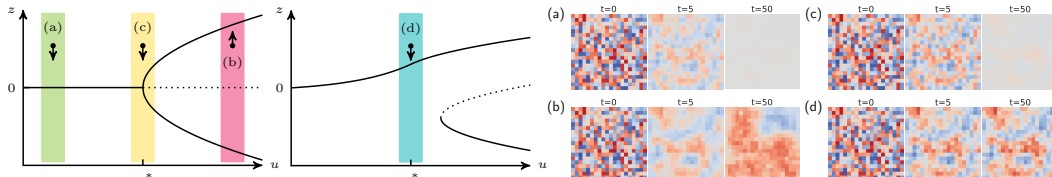

Figure 2: **Nonlinear opinion dynamics and dissensus. (Left)** The pitchfork bifurcation diagram illustrates a change in the number and stability of opinion states with the attention parameter $u$ (stable equilibria are illustrated with a solid line and unstable equilibria are illustrated with a dotted line). In the diagram, $z$ represents the weighted average of agent opinions, and $u^*$ represents the bifurcation point. When the input term $b = 0$ we have the pitchfork bifurcation (first from left), and when $b > 0$ we have its unfolding (second from left). **(Right)** The time evolution of agent opinions under the nonlinear opinion dynamics model depends on the initial weighted average of agent opinions $z$, the attention parameter $u$, and the weighted average of agent inputs $b$. Each subfigure corresponds to an initial condition on the left. In all cases, the initial $z$ is the same. **(a)** ($u < u^*$, $b = 0$). Agent opinions converge to a neutral consensus (i.e., perfect agreement) which is equivalent to oversmoothing. **(b)** ($u > u^*$, $b = 0$). Agent opinions converge to dissensus with low variance, $z$ is positive. **(c)** ($u = u^*$, $b = 0$). Agent opinions converge to a neutral consensus. **(d)** ($u = u^*$, $b > 0$). Agent opinions converge to dissensus with high variance, $z$ is positive.

$\mathbf{A}^{\mathrm{o}} = [a_{jl}^{\mathrm{o}}] \in \mathbb{R}^{N_{\mathrm{o}} \times N_{\mathrm{o}}}$, defines how correlated different options are. An opinion dynamics model on $\mathcal{M} = (\mathcal{G}^a, \mathcal{G}^o)$ describes the evolution of agent opinions in time.

**Opinion consensus.** Given a multi-agent system $\mathcal{M} = (\mathcal{G}^a, \mathcal{G}^o)$, a question of interest is whether the opinions of agents tend toward consensus (i.e., perfect agreement). The opinions of agents is said to reach consensus if and only if the opinions of all agents tend toward the same value as time tends to infinity (DeGroot, 1974), that is,

$$\lim_{t \to \infty} \|\mathbf{x}_i(t) - \mathbf{x}_j(t)\|_2^2 = 0, \text{ for all } e_{ij} \in \mathcal{E}. \tag{4}$$

**Linear opinion dynamics.** The linear opinion dynamics model introduced in DeGroot (1974) describes the discrete-time evolution of agent opinions by,

$$\mathbf{x}_i(t + 1) = \sum_{k=1}^{N_{\mathrm{a}}} a_{ik}^{\mathrm{a}} \mathbf{x}_k(t), \quad \sum_{k=1}^{N_{\mathrm{a}}} a_{ik}^{\mathrm{a}} = 1, \tag{5}$$

where $a_{ik}^{\mathrm{a}} \geq 0$ can be interpreted as the influence of agent $x_i$ on agent $x_k$, and the total influence of any agent sums to one. In this model, the option graph can be understood as uncorrelated, $a_{jl}^{\mathrm{o}} = 0$. The continuous-time analogue of Equation (5) is given by,

$$\dot{\mathbf{x}}_i(t) = -d_i \mathbf{x}_i(t) + \sum_{k=1}^{N_a} a_{ik}^{\mathrm{a}} \mathbf{x}_k(t), \tag{6}$$

where the total influence of agent $i$ is $d_i$ (Leonard et al., 2024). In linear opinion dynamics, consensus is reached for all initial conditions, and the consensus value is independent of the graph structure and linearly dependent on initial conditions (Leonard et al., 2024). The more general case of linear opinion dynamics with time-varying influence (i.e., time-dependent $a_{ik}^{\mathrm{a}}(t)$) can also be shown to converge to consensus (Moreau, 2005; Nedić & Liu, 2017; Fax & Murray, 2004; Blondel et al., 2005). Becchetti et al. (2020) further gives the upper bound of the convergence time to consensus for both the time-dependent and independent $a_{ik}^{\mathrm{a}}$.

**Nonlinear opinion dynamics.** The nonlinear opinion dynamics model (Leonard et al., 2024; Bizyaeva et al., 2022) describes the continuous-time evolution of agent $i$'s opinion about option $j$ by,

$$\dot{x}_{ij} = -d_{ij} x_{ij} + S\left( u_i \left( \alpha_{ij} x_{ij} + \sum_{\substack{k=1 \\ k \neq i}}^{N_{\mathrm{a}}} a_{ik}^{\mathrm{a}} x_{kj} + \sum_{\substack{l=1 \\ l \neq j}}^{N_{\mathrm{o}}} a_{jl}^{\mathrm{o}} x_{il} + \sum_{\substack{k=1 \\ k \neq i}}^{N_{\mathrm{a}}} \sum_{\substack{l=1 \\ l \neq j}}^{N_{\mathrm{o}}} a_{ik}^{\mathrm{a}} a_{jl}^{\mathrm{o}} x_{kj} \right) \right) + b_{ij}, \tag{7}$$

where $d_{ij} \geq 0$, $u_i > 0$, and $\alpha_{ij} \geq 0$ are parameters intrinsic to the agent, and $a_{jk}^{\mathrm{a}}$, $a_{jl}^{\mathrm{o}}$, and $b_{ij}$ are parameters extrinsic to the agent. The intrinsic parameter $d_{ij}$, the damping parameter, describes how

Table 1: **GNN-OD Analogy.** We describe our analogy between GNNs and opinion dynamics, relating graph structures and opinion dynamics, and notions of oversmoothing and opinion consensus.

| Graph neural networks | Opinion dynamics |
|---|---|
| Node | Agent |
| Edge | Communication link |
| Node feature dimension | Number of agent options |
| Value of node $i$ feature $j$ | Opinion of agent $i$ option $j$ |
| Oversmoothing | Opinion consensus |

resistant agent $i$ is to forming an opinion about option $j$; $u_i$, the attention parameter, represents the attentiveness of agent $i$ to the opinions of other agents; and $\alpha_{ij}$, the self-reinforcement parameter, defines how confident agent $i$ is in its opinion on option $j$. The extrinsic parameter $b_{ij}$, the input parameter, represents the impact of the environment on the agent $i$'s opinion about option $j$, and the saturating function $\mathcal{S}$ is selected so that $S(0) = 0$, $S'(0) = 1$, and $S'''(0) \neq 0$.

By modeling opinion formation as a nonlinear process, the nonlinear opinion dynamics model can capture opinion consensus and dissensus, offering greater expressivity compared to the models surveyed in Becchetti et al. (2020), which converge only to consensus states. The nonlinearity induces a bifurcation where the number and/or stability of equilibrium solutions changes (see Figure 2 left). Consensus results when all agents select exactly the same equilibrium value, dissensus results otherwise. A switch from consensus to dissensus can result from a change in the attention parameter $u$, this can be seen by comparing the dynamics at point (a) and (b) in Figure 2 (second from right), or from a change in the input parameter $b$, this can be seen by comparing the dynamics at point (c) and (d) in Figure 2 (first from right).

### 3.3 Graph Neural Network-Opinion Dynamics Analogy

As described in Section 3.1 and 3.2, GNNs and opinion dynamics models have several features which can be understood analogously. The nodes in a GNNs are analogous to the agents in an opinion dynamics model, the edges between nodes are analogous to communication links between agents, and layer-wise oversmoothing is analogous to opinion consensus in time (see Equations (3) and (4)). We summarize our GNN-opinion dynamics (GNN-OD) analogy in Table 1.

## 4 Oversmoothing and Opinion Consensus

In this section, we use our analogy to prove oversmoothing in several classes of GNNs, beginning with linear discrete-depth GNNs and then Laplacian-based continuous-depth GNNs. The utility of the GNN-OD analogy for understanding oversmoothing motivates its use in the design of new architectures (Section 5). All proofs are provided in Appendix A.

### 4.1 Linear discrete-depth GNNs

Linear discrete-depth GNNs (e.g., SGC (Wu et al., 2019) and DGC (Wang et al., 2021)) can be described using layer-wise transformations of the form,

$$\mathbf{X}^{l+1} = \mathbf{A}\mathbf{X}^l\mathbf{W}^l = \mathbf{D}\tilde{\mathbf{A}}\mathbf{X}^l\mathbf{W}^l = \tilde{\mathbf{A}}\mathbf{D}\mathbf{X}^l\mathbf{W}^l, \tag{8}$$

where adjacency matrix $\mathbf{A} = \mathbf{D}\tilde{\mathbf{A}}$ for some right stochastic matrix $\tilde{\mathbf{A}}$ and diagonal matrix $\mathbf{D}$ with $\mathbf{D}_{ii} = \sum_j \mathbf{A}_{ij}$; and the transformation matrix $\mathbf{W}^l$ is layer dependent. We can use our GNN-OD analogy to show this class of GNNs will exhibit oversmoothing for all inputs, and all input graphs.

**Lemma 4.1** (Linear dynamics oversmooth). *Any discrete-depth graph neural network with linear aggregation exhibits oversmoothing.*

Previous works that have shown oversmoothing in various subclasses of linear discrete-depth GNNs include Wu et al. (2022) which proves oversmoothing in convolutional GNNs, and Keriven (2022) which proves oversmoothing in discrete-depth GNNs with linear mean aggregation , and Scholkemper et al. (2024) which shows oversmoothing in linearized GNNs without residual connection and proper initialization.

## 4.2 LAPLACIAN-BASED CONTINUOUS-DEPTH GNNS

**Laplacian dynamics.** For a given graph, the graph Laplacian of is defined as $\mathbf{L} = \mathbf{A} - \mathbf{D}$ where $\mathbf{A}$ is the graph adjacency matrix and $\mathbf{D}$ is a diagonal matrix with entries $\mathbf{D}_{ii} = \sum_j \mathbf{A}_{ij}$. GNNs with Laplacian dynamics (e.g., GRAND-$\ell$ (Chamberlain et al., 2021a) and GraphCON-Tran (Rusch et al., 2022)), can be described using the dynamical equation,

$$\dot{\mathbf{X}}(t) = -\mathbf{L}\mathbf{X}(t). \tag{9}$$

We can use our GNN-OD analogy to show this class of GNNs will exhibit oversmoothing for all inputs, and all input graphs.

**Lemma 4.2** (Laplacian dynamics oversmooth). *Any continuous-depth graph neural network with Laplacian dynamics exhibits oversmoothing.*

Oversmoothing of continuous-depth GNNs with Laplacian dynamics and time-varying adjacency matrix (e.g., GRAND-$n\ell$ (Chamberlain et al., 2021a)), can be shown by analogy with linear opinion dynamics models with time-varying influence (see Section 3.2). Oversmoothing in GRAND-$\ell$ (Chamberlain et al., 2021a) was previously shown in Thorpe et al. (2022); Choi et al. (2023). Cai & Wang (2020) further discussed the bound on the decay rate of Dirichlet energy, quantifying the oversmoothing behavior in Laplacian dynamics.

**Laplacian dynamics with an external input.** Continuous-depth GNNs with Laplacian dynamics and an external input $\mathbf{B}(t)$ can be described using the dynamical equation,

$$\dot{\mathbf{X}}(t) = -\mathbf{L}\mathbf{X}(t) + \mathbf{B}(t). \tag{10}$$

**Lemma 4.3** (Laplacian dynamics with an external input oversmooth). *Any continuous-depth graph neural network with Laplacian dynamics and an external input will exhibit oversmoothing when the dynamics are linear.*

For works that design $\mathbf{B}(t)$ to address oversmoothing in linear models (Thorpe et al., 2022; Choi et al., 2023), this shows that oversmoothing persists. In order to structurally resolve oversmoothing, we turn to nonlinear inductive biases with more complicated and controllable stability behavior.

## 5 BEHAVIOR-INSPIRED MESSAGE PASSING NEURAL NETWORK

In this section, we describe our Behavior-Inspired Message Passing (BIMP) GNN which leverages nonlinear opinion dynamics as an inductive bias. Nonlinear opinion dynamics is more general than linear opinion dynamics, and can be designed to converge to dissensus for general inputs. We begin with model definition, then prove desirable properties like robustness to oversmoothing, well behaved gradients, and adaptability to homophilic and heterophilic datasets. Proofs are shown in Appendix A.

### 5.1 MODEL DEFINITION

Let $\mathcal{G} = (\mathcal{V}, \mathcal{E})$ be an input graph with $n$ nodes, where each node has a $d_{\text{in}}$-dimensional feature representation. BIMP applies a learnable encoder $\phi$, decoder $\psi$, and nonlinear opinion dynamics model to produce an output. The encoder is defined $\phi : \mathbb{R}^{d_{\text{in}}} \to \mathbb{R}^{N_o}$, the decoder is defined $\psi : \mathbb{R}^{N_o} \to \mathbb{R}^{d_{\text{class}}}$, and our dynamics are defined by the equation,

$$\dot{\mathbf{X}}(t) = -d\mathbf{X}(t) + \tanh\left[u\left(\alpha\mathbf{X}(t) + \mathbf{A}^{\text{a}}\mathbf{X}(t) + \mathbf{X}(t)\mathbf{A}^{\text{o}\top} + \mathbf{A}^{\text{a}}\mathbf{X}(t)\mathbf{A}^{\text{o}\top}\right)\right] + \mathbf{X}(0). \tag{11}$$

In our dynamics, the parameters $d$ and $\alpha$ are hyperparameters, the attention parameter $u = \frac{d}{\alpha+3}$, the initial condition $\mathbf{X}(0) = \phi(\mathbf{X}_{\text{in}})$, and the adjacency matrices of communication and option graphs, $\mathbf{A}^{\text{a}}$ and $\mathbf{A}^{\text{o}}$ respectively, are learned. The output of our model is given by $\mathbf{Y} = \psi(\mathbf{X}(T))$, where the terminal time $T$ is a hyperparameter.

#### 5.1.1 THE COMMUNICATION AND OPTION GRAPHS.

The communication and option graphs are designed to allow for theoretical analysis, and reduce computational expense. For a nonlinear opinion dynamics model of the form,

$$\dot{\mathbf{X}} = -d\mathbf{X} + \tanh\left[u(\alpha\mathbf{X} + \mathbf{A}^{\text{a}}\mathbf{X})\right] + \mathbf{B}, \tag{12}$$

the communication adjacency matrix $\mathbf{A}^{\mathrm{a}}$ must have a simple leading eigenvalue to admit analysis (Leonard et al., 2024). BIMP defined in Equation (11) can be written in the form of Equation (12) by combining the communication and option graphs into a single effective adjacency matrix.

**Definition 5.1** (Effective adjacency matrix). Given the adjacency matrices of the communication and option graphs, $\mathbf{A}^{\mathrm{a}}$ and $\mathbf{A}^{\mathrm{o}}$, the effective adjacency matrix $\tilde{\mathbf{A}}$ is defined,

$$\tilde{\mathbf{A}} = (\mathbf{A}^{\mathrm{o}} + \mathbf{I}) \otimes (\mathbf{A}^{\mathrm{a}} + \mathbf{I}). \tag{13}$$

**Lemma 5.2.** *The general form of nonlinear opinion dynamics (Equation (11)) can be written,*

$$\dot{\mathbf{x}} = -d\mathbf{x} + \tanh\left[u\Big((\alpha - 1)\mathbf{x} + \tilde{\mathbf{A}}\mathbf{x}\Big)\right] + \mathbf{b}, \tag{14}$$

*where $\tilde{\mathbf{A}}$ is the effective adjacency matrix, $\mathbf{x} = \mathrm{vec}(\mathbf{X})$ and $\mathbf{b} = \mathrm{vec}(\mathbf{B})$.*

In order to analyze the behavior of BIMP, the effective adjacency matrix must be constrained to have a simple leading eigenvalue. We enforce this condition by constraining the learned effective adjacency matrix to be right stochastic (the leading eigenvalue of right stochastic matrices is $\lambda_{\max} = 1$). Learning this matrix directly is computationally prohibitive (the size of the effective adjacency matrix is $\mathbb{R}^{N_{\mathrm{o}} N_{\mathrm{a}} \times N_{\mathrm{o}} N_{\mathrm{a}}}$). To relieve computational burden, we instead learn the communication and option graphs separately. The entries of $\mathbf{A}^{\mathrm{a}} = [a_{ik}^{\mathrm{a}}]$ and $\mathbf{A}^{\mathrm{o}} = [a_{jl}^{\mathrm{o}}]$ are defined using multi-head attention,

$$a_{ik}^{\mathrm{a}} = \mathrm{softmax}\left(\frac{(\mathbf{W}_K^{\mathrm{a}}\mathbf{x}_i)^{\top}\mathbf{W}_Q^{\mathrm{a}}\mathbf{x}_k}{d_k^{\mathrm{a}}}\right), \quad a_{jl}^{\mathrm{o}} = \mathrm{softmax}\left(\frac{(\mathbf{W}_K^{\mathrm{o}}\mathbf{x}_j^{\top})^{\top}\mathbf{W}_Q^{\mathrm{o}}\mathbf{x}_l^{\top}}{d_k^{\mathrm{o}}}\right), \tag{15}$$

where $\mathbf{W}_K^{\mathrm{a}}$, $\mathbf{W}_Q^{\mathrm{a}}$, $\mathbf{W}_K^{\mathrm{o}}$ and $\mathbf{W}_Q^{\mathrm{o}}$ are the key and query weight matrices for communication and option graphs. A useful consequence of this approach is that the leading eigenvalue $\lambda_{\max}^{\tilde{a}}$ of $\tilde{\mathbf{A}}$ is constant and does not need to be recomputed during training.

**Lemma 5.3.** *The leading eigenvalue of the effective adjacency matrix $\tilde{\mathbf{A}}$ is $\lambda_{\max}^{\tilde{a}} = 4$.*

## 5.2 Parameter selection

**The attention parameter $u$.** In the nonlinear opinion dynamics model, the attention parameter $u$ is the bifurcation parameter. Near the bifurcation point $u^*$ (i.e., the point where the number and/or stability of solutions change), the model is ultrasensitive to the input $\mathbf{B}$, and agents will quickly form an opinion (see Figure 2). We design BIMP to be ultrasensitive to the input by setting the value of the attention parameter $u$ to the bifurcation point of the attention-opinion bifurcation diagram.

**Lemma 5.4** (Bifurcation point $u^*$). *The bifurcation point $u^*$ of the attention-opinion bifurcation diagram is equal to $d/(\alpha - 1 + \lambda_{\max}^{\tilde{a}})$.*

From Lemma 5.4 and 5.3, the value of the attention parameter at the bifurcation point is $u = \frac{d}{\alpha + 3}$.

**The input parameter $\mathbf{B}$.** In the nonlinear opinion dynamics model, the input parameter $\mathbf{B}$ transforms the bifurcation diagram from a symmetric pitchfork bifurcation to an unfolded pitchfork bifurcation (see Figure 2). This is a form of selective ultrasensitivity where the directions of ultrasensitivity are determined by the structure of the communication graph (Bizyaeva et al., 2022; Leonard et al., 2024).

In BIMP with effective adjacency matrix $\tilde{\mathbf{A}}$ and attention parameter $u = \frac{d}{\alpha + 3}$, oversmoothing depends on the choice of input parameter. We select the input parameter $\mathbf{B}$, so that BIMP converges to an equilibrium for all initial opinions.

**Lemma 5.5** (BIMP converges to equilibrium). *A BIMP model with a constant input parameter $\mathbf{B}$ converges to an equilibrium.*

We can now understand the oversmoothing behavior of BIMP by analyzing its equilibrium behavior. When the input parameter $\mathbf{B}$ is nonzero, BIMP will not exhibit oversmoothing.

**Theorem 5.6** (Dissensus in BIMP). *BIMP will not exhibit oversmoothing when the input parameter $\mathbf{B}$ is time-independent with unique entries.*

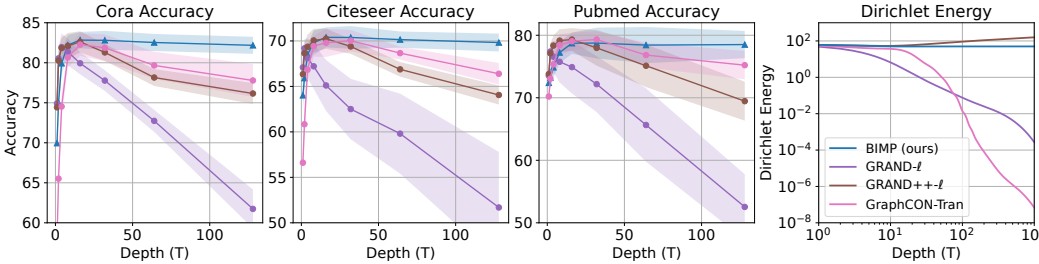

Figure 3: **Classification accuracy and Dirichlet energy.** BIMP is designed to learn node representations that resist oversmoothing even for very large depths. **(Left)** We compare the classification accuracy of BIMP to baseline models for architectures with $1, 2, 4, 8, 16, 32, 64$ and $128$ timesteps. Our BIMP model is stable out to 128 timesteps, while baseline performance deteriorates after 32 timesteps. **(Right)** We compare the Dirichlet energy of node features over a range of network depths. Contrary to baselines, the Dirichlet energy of BIMP remains stable even at very deep layers.

To ensure $\mathbf{B}$ satisfies the conditions of Theorem 5.6, we set $\mathbf{B} = \mathbf{X}(0)$. GRAND++ (Thorpe et al., 2022), GREAD (Choi et al., 2023) and KuramotoGNN (Nguyen et al., 2024) uses a similar strategy.

### 5.3 EMERGENT PROPERTIES

**Well behaved gradients.** The stability of neural network gradients impacts training efficiency and learning outcomes (Rusch et al., 2022; Nguyen et al., 2024; Pascanu et al., 2013; Awasthi et al., 2021; Arroyo et al., 2025). In GNNs, oversmoothing occurs when network gradients vanish (Rusch et al., 2022). In BIMP, we find that the structure of the nonlinear opinion dynamics inductive bias yields bounded gradients that do not vanish exponentially, even for very deep architectures.

**Theorem 5.7** (BIMP has well behaved gradients). *BIMP gradients are upper bounded and do not vanish exponentially.*

**Model adaptability.** In many GNNs, neighborhood aggregation can be interpreted as low-pass filtering (Nt & Maehara, 2019; Bo et al., 2021; Balcilar et al., 2021). A direct consequence is that these same GNNs will perform poorly on heterophilic datasets (i.e., datasets where edges in an input graphs connect dissimilar nodes). To address this issue, previous works have incorporated high-pass filters which have a sharpening effect (Han et al., 2023; Di Giovanni et al., 2022; Choi et al., 2023). In BIMP, we find that the nonlinear opinion dynamics inductive bias can be interpreted as a tunable filter. This becomes clear by writing the BIMP dynamics from Equation (14) in an alternative form,

$$\dot{\mathbf{x}} = -d\mathbf{x} + \tanh\left[u\big((\alpha-1)\underbrace{(\mathbf{x} - \tilde{\mathbf{A}}\mathbf{x})}_{\text{sharpening}} + \alpha\,\underbrace{\tilde{\mathbf{A}}\mathbf{x}}_{\text{smoothing}}\big)\right] + \mathbf{b}. \tag{16}$$

In this form, BIMP has a high pass filter when $\alpha > 1$ ($(\mathbf{x} - \tilde{\mathbf{A}}\mathbf{x})$ sharpens the features); and a low-pass filter when $\alpha \leq 1$ ($\tilde{\mathbf{A}}\mathbf{x}$ smooths the feature). By tuning $\alpha$, BIMP can be adapted to both homophilic and heterophilic datasets.

**Greater Expressivity.** GNN expressivity is constrained not only by oversmoothing (Oono & Suzuki, 2019; Li et al., 2018; Nt & Maehara, 2019) but also by limitations in model architecture (Xu et al., 2018; Alon & Yahav, 2020). BIMP incorporates nonlinearity and cross-dimensional feature mixing to enhance model expressivity, outperforming existing continuous-depth models (see Section 6).

**Theorem 5.8** (Expressive capacity of BIMP). *BIMP can model more diverse node feature representations than approaches whose dynamics are equivalent to linear opinion dynamics.*

## 6 EMPIRICAL ANALYSIS

In this section, we highlight the robustness of BIMP features to oversmoothing even in very deep architectures; and the classification accuracy of BIMP on homophilic, heterophilic, large graph and long-range graphs datasets.

### 6.1 PERFORMANCE AT LARGE DEPTHS

**Classification accuracy.** To understand the impact of depth on classification accuracy, we compare the performance of our BIMP model to GRAND-$\ell$ (Chamberlain et al., 2021a), GRAND++-$\ell$ (Thorpe

Table 2: **Classification accuracy on homophilic datasets.** Classification accuracy on the Cora, Citeseer, Pubmed, CoauthorCS, Computers, and Photo datasets are reported. Our BIMP model outperforms competitive baselines on all datasets. We highlight the best and second best accuracy.

| Dataset | Cora | Citeseer | Pubmed | CoauthorCS | Computers | Photo |
|---|---|---|---|---|---|---|
| **BIMP** | **83.19±1.13** | **71.09±1.40** | **80.16±2.03** | **92.48±0.26** | **84.73±0.61** | **92.90±0.44** |
| GRAND-$\ell$ | 82.20±1.45 | 69.89±1.48 | 78.19±1.88 | 90.23±0.91 | 82.93±0.56 | 91.93±0.39 |
| GRAND++-$\ell$ | 82.83±1.31 | 70.26±1.46 | 78.89±1.96 | 90.10±0.78 | 82.79±0.54 | 91.51±0.41 |
| KuramotoGNN | 81.16±1.61 | 70.40±1.02 | 78.69±1.91 | 91.05±0.56 | 80.06±1.60 | 92.77±0.42 |
| GraphCON-Tran | 82.80±1.34 | 69.60±1.16 | 78.85±1.53 | 90.30±0.74 | 82.76±0.58 | 91.78±0.50 |
| GAT | 79.76±1.50 | 67.70±1.63 | 76.88±2.08 | 89.51±0.54 | 81.73±1.89 | 89.12±1.60 |
| GCN | 80.76±2.04 | 67.54±1.98 | 77.04±1.78 | 90.98±0.42 | 82.02±1.87 | 90.37±1.38 |
| GraphSAGE | 79.37±1.70 | 67.31±1.63 | 75.52±2.19 | 90.62±0.42 | 76.42±7.60 | 88.71±2.68 |

et al., 2022), and GraphCON (Rusch et al., 2022) over a range of network depths (hyperparameters remain fixed across depths). We report classification performance in Figure 3 (left). BIMP performs comparably to baseline models for shallow depths ($l < 32$) and consistently outperforms all baseline models at greater depths. Experimental details are provided in Appendix D.1.1. We report additional comparison against discrete-depth GNNs and baselines amended with various oversmoothing mitigation techniques, including rewiring, normalization and skip connection, in Appendix D.1.1.

**Dirichlet energy.** To understand how oversmoothing evolves with depth, we compare the Dirichlet energy (see Equation (2)) of our BIMP model to baseline models over 1000 timesteps and plot the results in Figure 3 (right). The Dirichlet energy of our BIMP model is stable for all timesteps while the energy of baseline models diverges. Additional experimental details are provided in Appendix D.1.2. We report additional comparison against discrete-depth GNNs and baselines amended with various oversmoothing mitigation techniques in Appendix D.1.2.

## 6.2 CLASSIFICATION ACCURACY

**Homophilic datasets.** We report the classification performance of our BIMP model and competitive baselines (GRAND-$l$, GRAND++-$\ell$, KuramotoGNN (Nguyen et al., 2024), GraphCON, GAT (Veličković et al., 2017), GCN (Kipf & Welling, 2016), and GraphSAGE (Hamilton et al., 2017)) in Table 2, across 20 random initializations and 100 random train–validation–test splits, on the full datasets of Cora (McCallum et al., 2000), Citeseer (Sen et al., 2008), Pubmed (Namata et al., 2012), CoauthorCS (Shchur et al., 2018), Computers (Shchur et al., 2018), and Photo (Shchur et al., 2018). Furthermore, we report the performance of baselines amended with various oversmoothing mitigation techniques, including rewiring, layer-wise normalization and skip connection, in Table 4, Appendix D.2.1 and results on the standard Planetoid splits in Table 5. Our BIMP model outperforms all baseline and amended baseline models on all datasets.

**Heterophilic datasets.** We report the classification performance of our BIMP model and competitive baselines on the three small datasets, Texas, Wisconsin, Cornell (Craven et al., 1998), and three larger datasets Actor (Pei et al., 2020), Squirrel, Chameleon (Rozemberczki et al., 2021), across 100 random initializations and 10 standard splits, in Table 6, Appendix D.2.2. Our BIMP model outperforms all baselines on larger datasets and continuous-depth baselines on smaller datasets.

**Large graph.** We report the classification performances on the ogbn-arXiv (Hu et al., 2020) dataset with 20 random initialization on the standard split in Appendix D.2.3, Table 7, where BIMP outperforms baseline models, illustrating its scalability to large graphs.

Table 3: **Performance on LRGB benchmark.** We adopt baseline results reported in Gravina et al. (2025). Our BIMP outperforms all the continuous-depth models and on par with the SOTA result.

| Model | | Peptides-func (AP ↑) | Peptides-struct (MAE ↓) |
|---|---|---|---|
| GCN | | 59.30±0.23 | 0.3496±0.0013 |
| GIN | (Xu et al., 2018) | 54.98±0.79 | 0.3547±0.0045 |
| Transformer+LapPE | (Dwivedi & Bresson, 2020) | 63.26±1.26 | **0.2529±0.0016** |
| SAN+LapPE | (Kreuzer et al., 2021) | **63.84±1.21** | 0.2683±0.0043 |
| GRAND-$\ell$ | | 57.89±0.62 | 0.3418±0.0015 |
| GraphCON | | 60.22±0.68 | 0.2778±0.0018 |
| ADGN | (Gravina et al., 2022) | 59.75±0.44 | 0.2874±0.0021 |
| BIMP (ours) | | 63.62±1.07 | 0.2629±0.0027 |

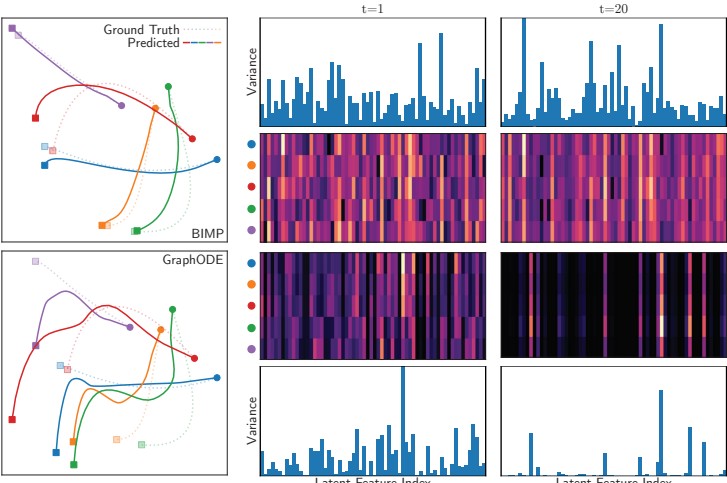

Figure 4: **Oversmoothing in trajectory extrapolation tasks**. We show the latent features at timestep 1 (●) and 20 (■) for both BIMP and GCN-based GraphODE. The latent features from GraphODE converges to a similar solution while BIMP maintains large variability.

**Long-range graphs and over-squashing.** Oversquashing is another fundamental limitation that restricts long-range information propagation in GNNs (Alon & Yahav, 2020; Topping et al., 2021). We report the prediction results on the datasets Peptides-func and Peptides-struct from Long Range Graph Benchmark (Dwivedi et al., 2022) in Table 3. BIMP outperforms all the continuous-depth models and performs on par with Graphormers which incur higher computational and memory costs, which shows that BIMP empirically mitigating oversquashing. Training detail is in Appendix D.2.4.

### 6.3 Additional Experiments

**Multi-agent Trajectory Extrapolation.** In our motivating experiment (Figure 1), we show that the GCN-based GraphODE tends to collapse trajectories, which degrades predictive accuracy and highlights the critical role of oversmoothing, especially in capturing long-term behavior. In contrast, BIMP avoids trajectory collapse and achieves superior predictive performance. Details and results are presented in Figure 7 and Appendix E.1.

Figure 4 visualizes the latent features predicted by BIMP and GraphODE at the initial ($t = 1$) and final step ($t = 20$). The top and bottom panels show the variance in each latent feature dimension, while the middle two panels display heatmaps of the respective 64-dimensional latent features for 5 balls. In GraphODE, oversmoothing becomes pronounced at $t = 20$, where most dimensions of the latent representation collapse to nearly identical values and variance approaches zero, leading the decoder to produce nearly identical predictions. In contrast, BIMP maintains meaningful discriminability in its latent features at $t = 20$.

**Empirical Evaluation.** We report the computational complexity of BIMP in Table 11 and Table 12, Appendix E.2, showing that it achieves comparable computational cost than both continuous-depth and discrete-depth baselines. We further ablate the nonlinearity function (Appendix E.3) and the inductive terms (Appendix E.4), observing performance changes that align well with our theoretical analysis. In addition, we conduct a sensitivity analysis to demonstrate that our hyperparameters damping $d$ and self-reinforcement $\alpha$ are generally robust across tasks (Figure 9, Appendix E.5).

### 7 Conclusion

In this paper, we propose an analogy between GNNs and opinion dynamics models, highlighting the equivalence between oversmoothing in GNNs and consensus in opinion dynamics. Through our analogy, we prove that several existing GNN algorithms are equivalent to linear opinion dynamics models which converges to consensus. Motivated by this, we introduce a novel class of continuous-depth GNNs called Behavior-inspired message passing (BIMP) which leverage the nonlinear opinion dynamics inductive bias, improving expressivity and guaranteeing dissensus. Experiments against recent baselines illustrate our model's competitive performance and robustness to oversmoothing.

**Limitations.** The nonlinear opinion dynamics inductive bias may introduce training instabilities at larger step sizes when using first order methods. If the step size is larger than $1/d$, each update will lead to a sign change. However, since the $d$ is a hyperparameter this is not a severe limitation.

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

## A PROOFS

### A.1 OVERSMOOTHING AND OPINION CONSENSUS

#### A.1.1 LEMMA 4.1: LINEAR DYNAMICS OVERSMOOTH

*Any discrete-depth graph neural network with linear aggregation exhibits oversmoothing.*

*Proof.* Without loss of generality, we consider the case where $\mathbf{A}$ is right stochastic. In this case, we can write the output of the $L$-th layer,

$$\mathbf{X}^L = \mathbf{A}\mathbf{X}^{L-1}\mathbf{W}^L = \mathbf{A}^{(L)}\mathbf{X}^0\mathbf{W}^0\cdots\mathbf{W}^L. \tag{17}$$

where $\mathbf{A}^{(n)}$ denotes the $n$-th power of matrix $\mathbf{A}$.

By our GNN-OD analogy, the quantity $\mathbf{A}^{(L)}\mathbf{X}^0$ tends toward consensus for all initial conditions. Since this cannot be changed by a linear transformation (e.g., the product $\mathbf{W}^0\cdots\mathbf{W}^L$), $\mathbf{X}^L$ will also tend toward consensus. Since oversmoothing is analogous to opinion consensus, any discrete-time GNN with linear aggregation will exhibit oversmoothing for all inputs. $\square$

#### A.1.2 LEMMA 4.2: LAPLACIAN DYNAMICS OVERSMOOTH

*Any continuous-depth graph neural network with Laplacian dynamics exhibits oversmoothing.*

*Proof.* Any GNN with Laplacian dynamics can be expressed in the form,

$$\dot{\mathbf{X}}(t) = -\mathbf{D}\mathbf{X}(t) + \mathbf{A}\mathbf{X}(t). \tag{18}$$

In this form, it is clear that the dynamics are equivalent to linear opinion dynamics (see Equation (6)). Since in a linear opinion dynamics model consensus is reached for all initial conditions, a GNN with Laplacian dynamics will also exhibit oversmoothing for all inputs. $\square$

#### A.1.3 LEMMA 4.3: LAPLACIAN DYNAMICS WITH AN EXTERNAL INPUT OVERSMOOTH

*Any continuous-depth graph neural network with Laplacian dynamics and an external input will exhibit oversmoothing when the dynamics are linear.*

*Proof.* We prove Lemma 4.3 by demonstrating potential oversmoothing behavior in two popular methods: GRAND++$\ell$ Thorpe et al. (2022) (in Lemma A.1), GREAD-F and GREAD-FB* Choi et al. (2023) (in Lemma A.2). The insight that the stability of linear dynamics is sensitive to the external input can be generalized to the analysis of other Laplacian dynamics with an external input. $\square$

**Lemma A.1** (Oversmoothing in GRAND++-$\ell$). *GRAND++-$\ell$ exhibits oversmoothing.*

*Proof.* In GRAND++-$\ell$, the layer-wise transformations is of the form,

$$\dot{\mathbf{X}} = -\mathbf{L}\mathbf{X} + \mathbf{B}, \tag{19}$$

where $\mathbf{B}$ is the fixed source term that depends on the initial state $\mathbf{X}(0)$. Defining the right eigenvector matrix $\mathbf{T} = [\mathbf{v}_1, \mathbf{v}_2, ..., \mathbf{v}_{N_a}]$ and left eigenvector matrix $\mathbf{T}^{-1} = [\mathbf{w}_1, \mathbf{w}_2, ..., \mathbf{w}_{N_a}]^\top$ for the graph Laplacian $\mathbf{L}$, we perform change of coordinates $\mathbf{Y} = [\mathbf{y}_1, \mathbf{y}_2, ..., \mathbf{y}_{N_a}]^\top = \mathbf{T}^{-1}\mathbf{X}$ such that

$$\mathbf{T}\dot{\mathbf{Y}} = -\mathbf{L}\mathbf{T}\mathbf{Y} + \mathbf{B}. \tag{20}$$

Decompose $\mathbf{L}$ by its right eigenvectors such that $\mathbf{L} = \mathbf{T}\mathbf{\Lambda}\mathbf{T}^{-1}$, Equation (20) further simplifies as

$$\mathbf{T}\dot{\mathbf{Y}} = -\mathbf{T}\mathbf{\Lambda}\mathbf{T}^{-1}\mathbf{T}\mathbf{Y} + \mathbf{B} = -\mathbf{T}\mathbf{\Lambda}\mathbf{Y} + \mathbf{B}. \tag{21}$$

Multiplying both sides with $\mathbf{T}^{-1}$ yields

$$\dot{\mathbf{Y}} = -\mathbf{\Lambda}\mathbf{Y} + \mathbf{T}^{-1}\mathbf{B}. \tag{22}$$

Since the eigenvalue matrix $\mathbf{\Lambda}$ is diagonal, Equation (22) can be decoupled into

$$\dot{\mathbf{y}}_i^\top = -\lambda_i \mathbf{y}_i^\top + \mathbf{w}_i^\top \mathbf{B}, \tag{23}$$

where $\lambda_i$ is the $i$-th eigenvalue of $\mathbf{\Lambda}$. Consider the case of $\lambda_i > 0$, the solution to the ODE becomes

$$\mathbf{y}_i^\top(t) = \frac{\mathbf{b}_i}{\lambda_i} + \mathbf{c}_i^\top e^{-\lambda_i t}, \tag{24}$$

where $\mathbf{b}_i = \mathbf{w}_i^\top \mathbf{B}$ and $\mathbf{c}_i$ are constant vectors. Consider the case where $\lambda_0 = 0$, the term $-\lambda_i \mathbf{y}_i^\top$ in Equation (23) becomes $\mathbf{0}$, hence the solution to the ODE becomes

$$\mathbf{y}_0^\top(t) = \mathbf{b}_0 t + \mathbf{c}_0^\top. \tag{25}$$

where $\mathbf{b}_0 = \mathbf{w}_0^\top \mathbf{B}$ and $\mathbf{c}_0$ are constant vectors. The solution in the original coordinate frame becomes

$$\mathbf{X}(t) = \sum_{\lambda_i > 0} \mathbf{v}_i \left( \frac{\mathbf{b}_i}{\lambda_i} + \mathbf{c}_i^\top e^{-\lambda_i t} \right) + \mathbf{v}_0 (\mathbf{b}_0 t + \mathbf{c}_0^\top), \tag{26}$$

where $\lambda_i$ and $\mathbf{v}_i$ denote the positive eigenvalues and corresponding Eigenvectors of $\mathbf{L}$; $\mathbf{b}_i = \mathbf{w}_i^\top \mathbf{B}$, where $\mathbf{w}_i$ are the left eigenvectors of $\mathbf{L}$. Particularly, $\mathbf{v}_0$ is an all-ones vector, i.e., $\mathbf{v}_0 = [1, 1, .., 1]^\top$, $\mathbf{b}_0 = \mathbf{w}_0^\top \mathbf{B}$, and $\mathbf{c}_0$, $\mathbf{c}_i$ are constant vectors. As $t \to \infty$, the exponential terms $\mathbf{c}_i^\top e^{-\lambda_i t}$ decays to zero and Equation (26) tends towards a linear system dominated by $\mathbf{v}_0$. Moreover, for large timescales, the difference in node features is relatively small resulting in reduced discriminability (another form of oversmoothing), and poorer network performance. □

**Lemma A.2** (Oversmoothing in GREAD-F and GREAD-FB*). *GREAD-F and GREAD-FB* exhibits oversmoothing.*

*Proof.* **GREAD-F.** In GREAD-F, the layer-wise transformations is of the form,

$$\dot{\mathbf{X}} = -\mathbf{L}\mathbf{X} + \mathbf{X} \odot (1 - \mathbf{X}), \tag{27}$$

where $\odot$ denotes the Hadamard product. For Laplacian $\mathbf{L}$, there exist a constant $C > 0$ such that

$$|[\mathbf{L}\mathbf{X}]_i| \leq C|\mathbf{X}_i|, \tag{28}$$

where $C$ depends on the maximum degree and largest edge weights and $\mathbf{X}_i$ denotes the $i$-th row of $\mathbf{X}$. Equation (27) can therefore be rewritten as

$$\dot{\mathbf{X}}_i = -[\mathbf{L}\mathbf{X}]_i + \mathbf{X}_i(1 - \mathbf{X}_i) \leq C|\mathbf{X}_i| + \mathbf{X}_i - \mathbf{X}_i^2. \tag{29}$$

Notably, when $\mathbf{X}_i < -C$, the RHS of Equation (29) is strictly smaller than zero. Since the dominant term $\mathbf{X}_i^2$ grows quadratically, the solution diverges to infinity

$$\lim_{t \to \infty} \mathbf{X}_i(t) = -\infty \quad \text{for all } i. \tag{30}$$

Therefore, as long as the maximum of $\mathbf{X}$ is smaller than some negative threshold $-C$, the entire system becomes monotonically decreasing, eliminating the possibility of equilibrium or steady-state convergence. While node values may diverge at different rates, the components with the fastest decay rate dominates the overall behavior. The remaining components which decays more slowly becomes negligible in relative magnitude. Therefore, all node features collapse and oversmoothing occurs.

**GREAD-FB$^*$.** In GREAD-FB$^*$, the layer-wise transformations is of the form,

$$\dot{\mathbf{X}} = -\alpha\mathbf{L}\mathbf{X} + \beta(\mathbf{L}\mathbf{X} + \mathbf{X}), \tag{31}$$

where $\alpha, \beta$ are trainable parameters to (de-)emphasize each term. As the external input is a linear transformation, the dynamics can be rewritten as

$$\dot{\mathbf{X}} = \underbrace{\Big((\beta - \alpha)\mathbf{L} + \beta\mathbf{I}\Big)}_{\tilde{\mathbf{L}}} \mathbf{X}. \tag{32}$$

Given the property of linear opinion dynamics, when $\alpha > \beta > 0$, the eigenvalue of $\tilde{\mathbf{L}}$ exists in range $[2\Delta(\beta - \alpha), \beta]$, where $\Delta$ is the maximum graph degree. In particular, the smallest eigenvalue of $\mathbf{L}$, which is zero, maps to the largest eigenvalue of $\lambda_{\tilde{\mathbf{L}}} = \beta$. This guarantees a global stable equilibrium cannot exist as at least one mode grows exponentially with rate $\beta$.

Meanwhile, as $t \to \infty$, the remaining components associated with smaller eigenvalues either increases slowly or decays to zero (depends on the sign of the eigenvalue), and thus becoming negligible in relative magnitude. Consequently, all node features are asymptotically dominated by the leading eigenvector, which is $[1, \cdots, 1]^\top$. This leads to a collapse in feature diversity and oversmoothing occurs. $\qquad\square$

## A.2 Behavior-Inspired Message Passing Neural Network

### A.2.1 Lemma 5.2

*The general form of nonlinear opinion dynamics (Equation (11)),*

$$\dot{\mathbf{X}}(t) = -d\mathbf{X}(t) + \tanh\left[u\left(\alpha\mathbf{X}(t) + \mathbf{A}^{\mathrm{a}}\mathbf{X}(t) + \mathbf{X}(t)\mathbf{A}^{\mathrm{o}\top} + \mathbf{A}^{\mathrm{a}}\mathbf{X}(t)\mathbf{A}^{\mathrm{o}\top}\right)\right] + \mathbf{X}(0),$$

*can be written,*

$$\dot{\mathbf{x}} = -d\mathbf{x} + \tanh\left[u\left((\alpha - 1)\mathbf{x} + \tilde{\mathbf{A}}\mathbf{x}\right)\right] + \mathbf{b}, \tag{33}$$

*where $\tilde{\mathbf{A}}$ is the effective adjacency matrix, $\mathbf{x} = \mathrm{vec}(\mathbf{X})$ and $\mathbf{b} = \mathrm{vec}(\mathbf{B})$.*

*Proof.* We first rewrite Equation (11) as

$$\dot{\mathbf{X}} = -d\mathbf{X} + \tanh\left[u\left((\alpha - 1)\mathbf{X} + (\mathbf{A}^{\mathrm{a}} + \mathbf{I})\mathbf{X}(\mathbf{A}^{\mathrm{o}\top} + \mathbf{I})\right)\right] + \mathbf{B}. \tag{34}$$

From here, we can write the matrix product $\mathbf{A}\mathbf{B}\mathbf{C}$ with $\mathbf{A} \in \mathbb{R}^{m \times m}$, $\mathbf{B} \in \mathbb{R}^{m \times n}$ and $\mathbf{C} \in \mathbb{R}^{n \times n}$, as $\mathbf{A}\mathbf{B}\mathbf{C} = \mathrm{vec}^{-1}\left[\left(\mathbf{C}^\top \otimes \mathbf{A}\right)\mathrm{vec}(\mathbf{B})\right]$, where vec denotes the vectorization operator Magnus & Neudecker (2019). This yields the following form

$$\dot{\mathbf{X}} = -d\mathbf{X} + \tanh\left[u\left((\alpha - 1)\mathbf{X} + \mathrm{vec}^{-1}(\tilde{\mathbf{A}}\,\mathrm{vec}(\mathbf{X}))\right)\right] + \mathbf{B}, \tag{35}$$

where $\tilde{\mathbf{A}} = (\mathbf{A}^{\mathrm{o}} + \mathbf{I}) \otimes (\mathbf{A}^{\mathrm{a}} + \mathbf{I})$ follows from Definition 5.1. Vectorizing both sides of yields

$$\dot{\mathbf{x}} = -d\mathbf{x} + \tanh\left[u\left((\alpha - 1)\mathbf{x} + \tilde{\mathbf{A}}\mathbf{x}\right)\right] + \mathbf{b}, \tag{36}$$

where $\mathbf{x} = \mathrm{vec}(\mathbf{X})$, and $\mathbf{b} = \mathrm{vec}(\mathbf{B})$. We obtain the general nonlinear opinion dynamics in the form of Equation (12). By vectorizing $\mathbf{X}$, each agent opinion on an option is treated as an individual agent-like entity, thereby reducing the original dynamics to the form where options are uncorrelated. $\qquad\square$

### A.2.2 LEMMA 5.3

*The leading eigenvalue of the effective adjacency matrix $\tilde{\mathbf{A}}$ is $\lambda_{\max}^{\tilde{a}} = 4$.*

*Proof.* Since both the communication adjacency matrix $\mathbf{A}^{\mathrm{a}}$ and the belief adjacency matrix $\mathbf{A}^{\mathrm{o}}$ are right stochastic, their leading eigenvalues are equal to one (i.e., $\lambda_{\max}^{\mathrm{a}} = \lambda_{\max}^{\mathrm{o}} = 1$). Since $\tilde{\mathbf{A}} = (\mathbf{A}^{\mathrm{o}} + \mathbf{I}) \otimes (\mathbf{A}^{\mathrm{a}} + \mathbf{I})$, its leading eigenvalue is equal to $\lambda_{\max}^{\tilde{a}} = (\lambda_{\max}^{\mathrm{a}} + 1)(\lambda_{\max}^{\mathrm{o}} + 1) = 4$. $\square$

### A.2.3 LEMMA 5.4: BIFURCATION POINT $u^*$

*The bifurcation point $\mathrm{u}^*$ of the attention-opinion bifurcation diagram is equal to $d/(\alpha - 1 + \lambda_{\max}^{\tilde{a}})$.*

*Proof.* The bifurcation point of the attention-opinion bifurcation diagram, i.e., the point where the number and/or stability of the solutions change, occurs when the input parameter $\mathbf{B}$ is equal to zero.

Following Leonard et al. (2024), we use a linear analysis to determine the bifurcation point of the attention-opinion bifurcation diagram. The linearization of the nonlinear opinion dynamics model (Equation (36)) is given by $\dot{\boldsymbol{\omega}} = J(\mathbf{x}_e)\,\boldsymbol{\omega}$, where $J(\mathbf{x}_e)$ is the Jacobian evaluated at the equilibrium $\mathbf{x}_e$, and $\boldsymbol{\omega} = \mathbf{x} - \mathbf{x}_e$. At the equilibrium $\mathbf{x}_e = \mathbf{0}$, the Jacobian is given by

$$\mathbf{J} = (-d + u(\alpha - 1))\mathbf{I} + u\tilde{\mathbf{A}}. \tag{37}$$

The eigenvalue of the Jacobian determines the equilibrium stability. Denoting the eigenvalue of $\tilde{\mathbf{A}}$, $\lambda^{\tilde{a}}$, the eigenvalue of the Jacobian can be expressed,

$$\lambda' = -d + u(\alpha - 1) + u\lambda^{\tilde{a}}. \tag{38}$$

The bifurcation point of the attention-opinion bifurcation diagram occurs when the dominant eigenvalue of the Jacobian is zero, reaching the upper bound for stability of the equilibrium $\mathbf{x}_e$. As $u$ continues to increases and the dominant eigenvalue becomes positive, the equilibrium $\mathbf{x}_e$ become unstable and a bifurcation emerges. Solving for the critical value of the attention yields $u^* = d/(\alpha - 1 + \lambda_{\max}^{\tilde{a}})$. $\square$

### A.2.4 LEMMA 5.5: BIMP CONVERGES TO EQUILIBRIUM

*A BIMP model with time-independent input parameter $\mathbf{B}$, converges to an equilibrium.*

*Proof.* Due to the monotonicity of our BIMP model, the opinions are guaranteed to converge to an equilibrium. Without loss of generality, consider the case where the graph is undirected and the system has only one option (i.e., $\mathbf{A}^{\mathrm{o}} = 0$ with no interrelationship between options),

$$\dot{\mathbf{X}} = -d\mathbf{X} + \tanh\left[u(\mathbf{A}^{\mathrm{a}}\mathbf{X} + \alpha\mathbf{X})\right] + \mathbf{B}. \tag{39}$$

Let $\mathbf{p}$ be an permutation matrix that re-index our system into block diagonal form

$$\hat{\mathbf{A}}^{\mathrm{a}} = \mathbf{P}\mathbf{A}^{\mathrm{a}}\mathbf{P}^{\top} = \begin{bmatrix} \mathbf{A}_{11} & 0 & 0 & 0 \\ 0 & \mathbf{A}_{22} & 0 & 0 \\ \vdots & \vdots & \ddots & 0 \\ 0 & 0 & \dots & \mathbf{A}_{nn} \end{bmatrix}, \quad \hat{\mathbf{X}} = \mathbf{P}\mathbf{X} = \begin{bmatrix} \mathbf{X}_1 \\ \mathbf{X}_2 \\ \vdots \\ \mathbf{X}_n \end{bmatrix}, \quad \hat{\mathbf{B}} = \mathbf{P}\mathbf{B} = \begin{bmatrix} \mathbf{B}_1 \\ \mathbf{B}_2 \\ \vdots \\ \mathbf{B}_n \end{bmatrix}, \tag{40}$$

where $\mathbf{A}_{nn}$ are irreducible blocks or zero matrices. Considering $\mathbf{p}$ is an orthonormal vector, $\mathbf{A}^{\mathrm{a}}$ and $\mathbf{X}$ can be expressed as

$$\mathbf{A}^{\mathrm{a}} = \mathbf{P}^{\top}\hat{\mathbf{A}}^{\mathrm{a}}\mathbf{P}, \quad \mathbf{X} = \mathbf{P}^{\top}\hat{\mathbf{X}}, \quad \mathbf{B} = \mathbf{P}^{\top}\hat{\mathbf{B}}. \tag{41}$$

Substituting $\mathbf{A}^{\mathrm{a}}$ and $\mathbf{X}$ with $\hat{\mathbf{A}}^{\mathrm{a}}$ and $\hat{\mathbf{X}}$ respectively in Equation (39) yields

$$\mathbf{P}^{\top}\dot{\hat{\mathbf{X}}} = -d\mathbf{P}^{\top}\hat{\mathbf{X}} + \tanh\left[u(\mathbf{P}^{\top}\hat{\mathbf{A}}^{\mathrm{a}}\mathbf{P}\mathbf{P}^{\top}\hat{\mathbf{X}} + \alpha\mathbf{P}^{\top}\hat{\mathbf{X}})\right] + \mathbf{P}^{\top}\hat{\mathbf{B}}, \tag{42}$$

$$= \mathbf{P}^\top(-d\hat{\mathbf{X}}) + \mathbf{P}^\top \tanh\left[u(\hat{\mathbf{A}}^{\mathrm{a}}\hat{\mathbf{X}} + \alpha\hat{\mathbf{X}})\right] + \mathbf{P}^\top\hat{\mathbf{B}}, \tag{43}$$

and multiplying $\mathbf{P}$ on both sides

$$\dot{\hat{\mathbf{X}}} = -d\hat{\mathbf{X}} + \tanh\left[u(\hat{\mathbf{A}}^{\mathrm{a}}\hat{\mathbf{X}} + \alpha\hat{\mathbf{X}})\right] + \hat{\mathbf{B}}. \tag{44}$$

Leveraging the block diagonal form, Equation (44) can be decoupled into

$$\dot{\mathbf{X}}_m = -d\mathbf{X}_m + \tanh\left[u(\mathbf{A}_{mm}\mathbf{X}_m + \alpha\mathbf{X}_m)\right] + \mathbf{B}_m. \tag{45}$$

where the convergence of each subsystem $\dot{\mathbf{X}}_m = f_m(\mathbf{X}_m), 1 \le m \le n$ can be examined individually. The Jacobian of subsystem $\dot{\mathbf{X}}_m = f_m(\mathbf{X}_m)$ is defined as

$$\mathbf{J}_m(\mathbf{X}_m) = \frac{\partial f_m(\mathbf{X}_m)}{\partial \mathbf{X}_m} = -d\mathbf{I} + \mathbf{1}\mathrm{vec}\left(\mathrm{sech}^2\left(u(\mathbf{A}_{mm}\mathbf{X}_m + \alpha\mathbf{X}_m)\right)\right)^\top \circ \left[u((\mathbf{A}_{mm} \otimes \mathbf{I}) + \alpha\mathbf{I})\right], \tag{46}$$

where $\mathbf{I}$ is the identity matrix, $\mathbf{1}$ the all-ones vector and vec is the vectorization. $\circ$ is Hadamard product. Each subsystem and their associated Jacobian satisfies

- **Cooperative.** Since $\mathrm{sech} \in (0, 1]$, $\mathbf{A}_{mm}$ are positive matrices (as it is the output of a softmax function) and $\alpha \ge 0$, $\mathbf{J}_m(\mathbf{X}_m)$ is a Metzler matrix in which all the off-diagonal elements are non-negative. This implies that $\mathbf{J}_m(\mathbf{X}_m)$ is *cooperative* and the subsystem $\dot{\mathbf{X}}_m = f_m(\mathbf{X}_m)$ is monotone Hirsch (1985).

- **Irreducible.** As $\mathbf{A}_{mm}$ is constructed to be irreducible, $((\mathbf{A}_{mm}\otimes\mathbf{I})+\alpha\mathbf{I})$ remains irreducible and hence the Jacobian $\mathbf{J}_m(\mathbf{X}_m)$ is *irreducible*.

- **Compact closure.** The existence of the damping term $d$ in subsystem $\dot{\mathbf{X}}_m = f_m(\mathbf{X}_m)$ ensure the forward orbit has compact closure (i.e., bounded).

If the Jacobian for a continuous dynamical system $\dot{x} = f(x, t)$ is cooperative and irreducible, then it approaches the equilibrium for almost every point $x$ whose forward orbit has compact closure Hirsch (1985). Since the Jacobian $\mathbf{J}_m(\mathbf{X}_m)$ for each subsystem $\dot{\mathbf{X}}_m = f_m(\mathbf{X}_m)$ satisfies this condition, almost every state $\mathbf{X}_m$ approaches the equilibrium set. Therefore the dynamical system defined in Equation (39) converges to an equilibrium set. As all trajectories tend towards the equilibrium solution, analyzing the equilibrium behavior is sufficient to understand the underlying dynamics of our BIMP model.

If there are more than one option in the system (i.e, $\mathbf{A}^{\circ} \ne [0]$), the vectorized system defined in Equation (36) can be shown analogously to converge to its equilibrium set. $\qquad\square$

### A.2.5 Theorem 5.6: Dissensus in BIMP

> *BIMP will not exhibit oversmoothing when the input parameter $\mathbf{B}$ is time-independent with unique entries.*

*Proof.* Without loss of generality, consider the case where the graph is undirected and the system has only one option (i.e., $\mathbf{A}^{\circ} = \mathbf{0}$). By Definition 5.1, the effective adjacency matrix becomes

$$\tilde{\mathbf{A}} = 1 \otimes (\mathbf{A}^{\mathrm{a}} + \mathbf{I}) = \mathbf{A}^{\mathrm{a}} + \mathbf{I}. \tag{47}$$

Consider that $\mathbf{x} = [x_1, x_2, ..., x_{N_a}]^\top$, we can decouple the dynamical equation of $x_i$ from Equation (36) such that

$$\dot{x}_i = -dx_i + \tanh(u(\tilde{\alpha}x_i + \tilde{\mathbf{a}}_i\mathbf{x})) + b_i, \tag{48}$$

where $\tilde{\mathbf{a}}_i$ is the $i$-th row of $\tilde{\mathbf{A}}$ and $b_i$ is the $i$-th element of $\mathbf{b}$. Assume $\mathbf{x}$ converges to consensus such that $x_1 = x_2 = \ldots = x_{N_a} = \bar{x}$. For any pair $x_m$ and $x_n, m \ne n$ with corresponding input $b_m \ne b_n$, their dynamical equations are

$$\dot{x}_m = -dx_m + \tanh(u(\tilde{\alpha}x_m + \tilde{\mathbf{a}}_m\mathbf{x})) + b_m, \tag{49}$$

$$\dot{x}_n = -dx_n + \tanh(u(\tilde{\alpha}x_n + \tilde{\mathbf{a}}_n\mathbf{x})) + b_n. \tag{50}$$

We observe that

$$-dx_m = -dx_n = -d\bar{x}, \tag{51}$$
$$-\tilde{\alpha}x_m = -\tilde{\alpha}x_n = -\tilde{\alpha}\bar{x}, \tag{52}$$

and

$$\tilde{\mathbf{a}}_m\mathbf{x} = \tilde{\mathbf{a}}_n\mathbf{x} = 2\bar{x}, \tag{53}$$

due to $\tilde{\mathbf{a}}_m$ and $\tilde{\mathbf{a}}_n$ being right stochastic. However, since $b_m \neq b_n$, the right hand side of Equation (49) and Equation (50) cannot be 0 at the same time. Therefore, by contradiction, consensus cannot be the equilibrium for BIMP if $b_m \neq b_n$. If $\mathbf{b}$ has unique elements, the equilibrium of the system forms dissensus and avoids oversmoothing.

If there are more than one option in the system (i.e, $\mathbf{A}^o \neq \mathbf{0}$), formation of dissensus is still possible since $\mathbf{A}^o$ is also right stochastic. $\qquad\square$

### A.2.6 THEOREM 5.7: BIMP HAS WELL BEHAVED GRADIENTS

*BIMP gradients are upper bounded and do not vanish exponentially.*

*Proof.* For simplicity, consider the forward Euler method for integration of the dynamics defined by Equation (11) such that

$$\mathbf{X}^t = \mathbf{X}^{t-1} + \Delta t \dot{\mathbf{X}}^{t-1}, \tag{54}$$
$$\mathbf{X}^0 = \phi(\mathbf{X}_{\text{in}}) = \mathbf{W}\mathbf{X}_{\text{in}}, \tag{55}$$

where $\Delta t$ is the numerical integration step size, $\mathbf{X}^t \in \mathbb{R}^{N_a \times N_o}$ are the features at time $t \in [\Delta t, 2\Delta t, \ldots, M\Delta t]$, and $\mathbf{X}_{\text{in}} \in \mathbb{R}^{N_a \times f}$ are the input features. For the simplicity, we assume a linear encoder $\phi$ parameterized by learnable weights $\mathbf{W}$. Similar to existing continuous depth GNNs, the total steps of ODE integrations $M$ is interpreted as the number of layers of a model. Consider a node classification task using BIMP subject to mean squared error loss

$$\mathcal{L}(\mathbf{W}) = \frac{1}{2N_a N_o} \sum_{i=1}^{N_a} \sum_{j=1}^{N_o} (x_{ij}^M - \hat{x}_{ij})^2, \tag{56}$$

where $x_{ij}^M$ is an element of the learned features $\mathbf{X}^M$ at layer $M$ and $\hat{x}_{ij}$ is an element of the ground truth $\hat{\mathbf{X}}$. Consider all intermediate layers where $t \in [\Delta t, 2\Delta t, \ldots, M\Delta t]$, the gradient descent equation can be constructed as

$$\frac{\partial \mathcal{L}}{\partial \mathbf{W}} = \frac{\partial \mathcal{L}}{\partial \mathbf{X}^M} \frac{\partial \mathbf{X}^M}{\partial \mathbf{X}^1} \frac{\partial \mathbf{X}^1}{\partial \mathbf{X}^0} \frac{\partial \mathbf{X}^0}{\partial \mathbf{W}}, \tag{57}$$

where

$$\frac{\partial \mathbf{X}^M}{\partial \mathbf{X}^1} = \prod_{t=2}^{M} \frac{\partial \mathbf{X}^t}{\partial \mathbf{X}^{t-1}}. \tag{58}$$

With increasing depth (i.e, $M \to \infty$), this repeated multiplication leads to gradient exploding (vanishing) when the components $\partial \mathbf{X}^t / \partial x^{t-1} > \mathbf{I}$ ($\partial \mathbf{X}^t / \partial x^{t-1} < \mathbf{I}$). The BIMP model provides an upper and lower bound on gradients in Lemma A.3 and A.4 to guarantee exploding or vanish gradients cannot occur. $\qquad\square$

**Lemma A.3.** *BIMP gradients are upper bounded when the step-size $\Delta t \ll 1$ and damping term $d < 1/\Delta t$.*

*Proof.* Consider integrating BIMP with the forward Euler scheme defined in Equation (54) and (55) with fixed hyper parameters $\tilde{\alpha} = \alpha - 1$ and $u = d/\tilde{\alpha}+4$,

$$\mathbf{X}^t = \mathbf{X}^{t-1} + \Delta t \left( -d\mathbf{X}^{t-1} + \tanh\left[ u\left( \tilde{\alpha}\mathbf{X}^{t-1} + (\mathbf{A}^a + \mathbf{I})\mathbf{X}^{t-1}(\mathbf{A}^{o\top} + \mathbf{I}) \right) \right] + \mathbf{X}^0 \right) \tag{59}$$

$$= (1 - d\Delta t)\mathbf{X}^{t-1} + \Delta t \tanh\left[u\left(\tilde{\alpha}\mathbf{X}^{t-1} + \left(\mathbf{A}^{\mathrm{a}} + \mathbf{I}\right)\mathbf{X}^{t-1}\left(\mathbf{A}^{\circ\top} + \mathbf{I}\right)\right)\right] + \Delta t\mathbf{X}^0, \quad (60)$$

with initial feature embedding

$$\mathbf{X}^0 = \phi(\mathbf{X}_{\mathrm{in}}) = \mathbf{X}_{\mathrm{in}}\mathbf{W}. \tag{61}$$

Vectorizing Equation (60) and (61) yields

$$\mathbf{x}^t = (1 - d\Delta t)\mathbf{x}^{t-1} + \Delta t \tanh\left[u\left(\tilde{\alpha}\mathbf{I} + \tilde{\mathbf{A}}\right)\mathbf{x}^{t-1}\right] + \Delta t\mathbf{x}^0, \tag{62}$$

$$\mathbf{x}^0 = \tilde{\mathbf{W}}\mathbf{x}_{\mathrm{in}}, \tag{63}$$

where $\tilde{\mathbf{A}} = (\mathbf{A}^{\circ} + \mathbf{I}) \otimes (\mathbf{A}^{\mathrm{a}} + \mathbf{I})$, $\tilde{\mathbf{W}} = \mathbf{W}^\top \otimes \mathbf{I}_{N_a}$, and $\mathbf{x}^t = [x_1^t, x_2^t, \ldots, x_{N_a \times N_o}^t]^\top$. Reformulating gradient calculation in Equation (57) subject to loss function defined in Equation (56) with respect to the vectorized variables gives

$$\frac{\partial \mathcal{L}}{\partial \tilde{\mathbf{W}}} = \frac{\partial \mathcal{L}}{\partial \mathbf{x}^M} \frac{\partial \mathbf{x}^M}{\partial \mathbf{x}^1} \frac{\partial \mathbf{x}^1}{\partial \mathbf{x}^0} \frac{\partial \mathbf{x}^0}{\partial \tilde{\mathbf{W}}}, \tag{64}$$

$$\frac{\partial \mathbf{x}^M}{\partial \mathbf{x}^1} = \prod_{t=2}^{M} \frac{\partial \mathbf{x}^t}{\partial \mathbf{x}^{t-1}}, \tag{65}$$

where the upper bound for $\left\|\frac{\partial \mathbf{x}^M}{\partial \mathbf{x}^1}\right\|_\infty$, $\left\|\frac{\partial \mathcal{L}}{\partial \mathbf{x}^M}\right\|_\infty$, $\left\|\frac{\partial \mathbf{x}^1}{\partial \mathbf{x}^0}\right\|_\infty$ and $\left\|\frac{\partial \mathbf{x}^0}{\partial \tilde{\mathbf{W}}}\right\|_\infty$ can be found individually and are summarized in Equation (78), (84), (86), and (87) respectively.

Consider the first term $\left\|\frac{\partial \mathbf{x}^M}{\partial \mathbf{x}^1}\right\|_\infty$ and recalling that

$$\frac{\partial \mathbf{x}^M}{\partial \mathbf{x}^1} = \prod_{t=2}^{M} \frac{\partial \mathbf{x}^t}{\partial \mathbf{x}^{t-1}}. \tag{66}$$

By inspecting each term $\frac{\partial \mathbf{x}^t}{\partial \mathbf{x}^{t-1}}$, it follows that

$$\frac{\partial \mathbf{x}^t}{\partial \mathbf{x}^{t-1}} = (1 - d\Delta t)\mathbf{I} + \Delta t\mathbf{1}\left[\mathrm{sech}^2\left(u(\tilde{\alpha}\mathbf{I} + \tilde{\mathbf{A}})\mathbf{x}^{t-1}\right)\right]^\top \circ \left(u(\tilde{\alpha}\mathbf{I} + \tilde{\mathbf{A}})\right), \tag{67}$$

where $\mathbf{1}\left[\mathrm{sech}^2\left(u(\tilde{\alpha}\mathbf{I} + \tilde{\mathbf{A}})\mathbf{x}^{t-1}\right)\right]^\top$ represents a matrix repeating the vector $\mathrm{sech}^2\left(u(\tilde{\alpha}\mathbf{I} + \tilde{\mathbf{A}})\mathbf{x}^{t-1}\right)$ along the row dimension. $\circ$ is the Hadamard product. As $\mathrm{sech}(\cdot) \in (0, 1]$, we can leverage the triangle identity to obtain an upper bound

$$\left\|\frac{\partial \mathbf{x}^t}{\partial \mathbf{x}^{t-1}}\right\|_\infty \leq \left\|(1 - d\Delta t)\mathbf{I} + u\Delta t(\tilde{\alpha}\mathbf{I} + \tilde{\mathbf{A}})\right\|_\infty \tag{68}$$

$$\leq \|(1 - d\Delta t)\mathbf{I}\|_\infty + u\Delta t\|\tilde{\alpha}\|_\infty + u\Delta t\left\|\tilde{\mathbf{A}}\right\|_\infty. \tag{69}$$

Since $u = {}^d\!/\!_{\alpha+3}, \tilde{\alpha} = \alpha - 1, d > 0, \alpha \geq 0$, it follows that

$$u\Delta t\|\tilde{\alpha}\|_\infty = \frac{d}{\alpha + 3}\Delta t\|\alpha - 1\| < d\Delta t. \tag{70}$$

Since $\tilde{\mathbf{A}} = (\mathbf{A}^{\circ} + \mathbf{I}) \otimes (\mathbf{A}^{\mathrm{a}} + \mathbf{I})$ from Definition 5.1 and given that $\mathbf{A}^a$ and $\mathbf{A}^o$ are right stochastic, it follows that

$$\left\|\tilde{\mathbf{A}}\right\|_\infty < 4. \tag{71}$$

Therefore, Equation (69) can be further bounded by Equation (70) and (71) as

$$\left\|\frac{\partial \mathbf{x}^t}{\partial \mathbf{x}^{t-1}}\right\|_\infty \leq \|(1 - d\Delta t)\mathbf{I}\|_\infty + d\Delta t + 4u\Delta t, \tag{72}$$

$$< (1 - d\Delta t) + d\Delta t + 4u\Delta t, \tag{73}$$

$$< 1 + 4u\Delta t. \tag{74}$$

Since we assume $\Delta t \ll 1$, it follows that

$$(1 + 4u\Delta t)^{M-1} = 1 + 4(M-1)u\Delta t + \mathcal{O}(\Delta t^2), \tag{75}$$

$$\approx 1 + 4(M-1)u\Delta t, \tag{76}$$

$$< 1 + 4Mu\Delta t. \tag{77}$$

Finally, the term is upper bounded as

$$\left\|\frac{\partial \mathbf{x}^M}{\partial \mathbf{x}^1}\right\|_\infty \leq 1 + 4Mu\Delta t. \tag{78}$$

Consider the second term $\left\|\frac{\partial \mathcal{L}}{\partial \mathbf{x}^M}\right\|_\infty$

$$\frac{\partial \mathcal{L}}{\partial \mathbf{x}^M} = \frac{1}{N_a N_o} \operatorname{diag}(\mathbf{x}^M - \hat{\mathbf{x}}), \tag{79}$$

where $\operatorname{diag}(\mathbf{x}^M - \hat{\mathbf{x}})$ is a diagonal matrix with vector entry $\mathbf{x}^M - \hat{\mathbf{x}}$ on the diagonal and $\hat{\mathbf{x}} = \operatorname{vec}(\hat{\mathbf{X}})$. Taking the absolute value of Equation (62) and recalling $\tanh(\cdot) \in (-1, 1)$, it follows that

$$|x_i^M| \leq (1 - d\Delta t)|x_i^{M-1}| + (1 + |x_i^0|)\Delta t. \tag{80}$$

Therefore, recursively it can be shown that

$$|x_i^M| \leq (1 - d\Delta t)^M |x_i^0| + \left(\sum_{p=0}^{M-1} (1 - d\Delta t)^p\right)(1 + |x_i^0|)\Delta t, \tag{81}$$

$$\leq |x_i^0| + M(1 + |x_i^0|)\Delta t. \tag{82}$$

Taking $\infty$-norm of Equation (79) yields

$$\left\|\frac{\partial \mathcal{L}}{\partial \mathbf{x}^M}\right\|_\infty \leq \frac{1}{N_a N_o}(\|\mathbf{x}^M\|_\infty + \|\hat{\mathbf{x}}\|_\infty). \tag{83}$$

Substituting Equation (82) into Equation (83) result in the upper bound

$$\left\|\frac{\partial \mathcal{L}}{\partial \mathbf{x}^M}\right\|_\infty \leq \frac{1}{N_a N_o}\left(M\Delta t + (1 + M\Delta t)\|\mathbf{x}^0\|_\infty + \|\hat{\mathbf{x}}\|_\infty\right). \tag{84}$$

Consider the third term $\left\|\frac{\partial \mathbf{x}^1}{\partial \mathbf{x}^0}\right\|_\infty$. Since the input term $\mathbf{x}^0$ contributes to the differential defined in Equation (60), it follows that the upper bound can be derived as

$$\frac{\partial \mathbf{x}^1}{\partial \mathbf{x}^0} = \left(1 - (d-1)\Delta t\right)\mathbf{I} + \Delta t \mathbf{1}\left[\operatorname{sech}^2\left(u(\tilde{\alpha}\mathbf{I} + \tilde{\mathbf{A}})\mathbf{x}^0\right)\right]^\top \circ \left(u(\tilde{\alpha}\mathbf{I} + \tilde{\mathbf{A}})\right), \tag{85}$$

$$\left\|\frac{\partial \mathbf{x}^1}{\partial \mathbf{x}^0}\right\|_\infty < 1 + (4u + 1)\Delta t. \tag{86}$$

Consider the fourth term $\left\|\frac{\partial \mathbf{x}^0}{\partial \tilde{\mathbf{W}}}\right\|_\infty$, the upper bound can be defined as

$$\left\|\frac{\partial \mathbf{x}^0}{\partial \tilde{\mathbf{W}}}\right\|_\infty = \|\mathbf{x}_{\text{in}}\|_\infty. \tag{87}$$

Combining Equation (78), (84), (86), and (87), it follows that the upper bound for gradient calculations of BIMP is

$$\left\|\frac{\partial \mathcal{L}}{\partial \tilde{\mathbf{W}}}\right\|_\infty < \frac{1}{N_a N_o}\left(M\Delta t + (1 + M\Delta t)\|\mathbf{x}^0\|_\infty + \|\hat{\mathbf{x}}\|_\infty\right)\left(1 + 4Mu\Delta t\right)\left(1 + (4u+1)\Delta t\right)\|\mathbf{x}_{\text{in}}\|_\infty. \tag{88}$$

By designing hyperparameters

$$\beta = M\Delta t, \tag{89}$$

$$\gamma = (1 + 4Mu\Delta t)\left(1 + (4u + 1)\Delta t\right), \tag{90}$$

the upper bound defined in Equation (88) can be simplified as

$$\left\|\frac{\partial \mathcal{L}}{\partial \tilde{\mathbf{W}}}\right\|_\infty < \frac{1}{N_a N_o}\left(\beta + (1 + \beta)\|\mathbf{x}^0\|_\infty + \|\hat{\mathbf{x}}\|_\infty\right)\gamma\,\|\mathbf{x}_{\text{in}}\|_\infty. \tag{91}$$

Consider that $\mathbf{W}$ and $\tilde{\mathbf{W}}$ have the same elements, $\|\frac{\partial \mathcal{L}}{\partial \mathbf{W}}\|_\infty = \|\frac{\partial \mathcal{L}}{\partial \tilde{\mathbf{W}}}\|_\infty$ and therefore

$$\left\|\frac{\partial \mathcal{L}}{\partial \mathbf{W}}\right\|_\infty < \frac{1}{N_a N_o}\left(\beta + (1 + \beta)\|\mathbf{x}^0\|_\infty + \|\hat{\mathbf{x}}\|_\infty\right)\gamma\,\|\mathbf{x}_{\text{in}}\|_\infty. \tag{92}$$

which indicates that the gradients are upper bounded regardless of network depth and avoids exploding gradients. $\qquad\square$

**Lemma A.4.** *BIMP gradients will not vanish exponentially when the step-size $\Delta t \ll 1$ and the damping term $d < {}^1/_{\Delta t}$.*

*Proof.* From Equation (57), the terms $\frac{\partial \mathbf{X}^M}{\partial \mathbf{X}^1}$, $\frac{\partial \mathbf{X}^1}{\partial \mathbf{X}^0}$, $\frac{\partial \mathcal{L}}{\partial \mathbf{X}^M}$, and $\frac{\partial \mathbf{X}^0}{\partial \mathbf{W}}$ can be individually reformulated as a recursive summation operation and are summarized in Equation (93), (95), (99), and (100) respectively.

Consider the term $\frac{\partial \mathbf{X}^M}{\partial \mathbf{X}^1}$, which can be expressed as

$$\frac{\partial \mathbf{X}^t}{\partial \mathbf{X}^{t-1}} = \mathbf{I} + \Delta t \mathbf{E}_{t-1}, \tag{93}$$

where

$$\mathbf{E}_{t-1} = -d\mathbf{I} + \mathbf{1}\left[\operatorname{sech}^2\left(u(\tilde{\alpha}\mathbf{X}^{t-1} + (\mathbf{A}^a + \mathbf{I})\mathbf{X}^{t-1}(\mathbf{A}^{o\top} + \mathbf{I}))\right)\right]^\top$$
$$\circ\, u\left[\tilde{\alpha}\mathbf{I} + \left(\mathbf{I} \otimes (\mathbf{A}^o + \mathbf{I})\right)\left((\mathbf{A}^a + \mathbf{I}) \otimes \mathbf{I}\right)\right]. \tag{94}$$

Consider the term $\frac{\partial \mathbf{X}^1}{\partial \mathbf{X}^0}$, which can be reformulated as

$$\frac{\partial \mathbf{X}^1}{\partial \mathbf{X}^0} = \mathbf{I} + \Delta t \mathbf{E}_0, \tag{95}$$

where

$$\mathbf{E}_0 = (1 - d)\mathbf{I} + \mathbf{1}\operatorname{vec}\left(\operatorname{sech}^2\left(u(\tilde{\alpha}\mathbf{X}^{t-1} + (\mathbf{A}^a + \mathbf{I})\mathbf{X}^{t-1}(\mathbf{A}^{o\top} + \mathbf{I}))\right)\right)^\top$$
$$\circ\, u\left[\tilde{\alpha}\mathbf{I} + \left(\mathbf{I} \otimes (\mathbf{A}^o + \mathbf{I})\right)\left((\mathbf{A}^a + \mathbf{I}) \otimes \mathbf{I}\right)\right]. \tag{96}$$

Combining the previous two terms, it follows that

$$\frac{\partial \mathbf{X}^M}{\partial \mathbf{X}^0} = \left(\mathbf{I} + \Delta t \mathbf{E}_{M-1}\right)\left(\mathbf{I} + \Delta t \mathbf{E}_{M-2}\right)...\left(\mathbf{I} + \Delta t \mathbf{E}_0\right), \tag{97}$$

$$= \mathbf{I}_n + \Delta t\left(\mathbf{E}_0 + \sum_{t=1}^{M-1}\mathbf{E}_t\right) + \mathcal{O}(\Delta t^2). \tag{98}$$

Consider the term $\frac{\partial \mathcal{L}}{\partial \mathbf{X}^M}$

$$\frac{\partial \mathcal{L}}{\partial \mathbf{X}^M} = \frac{1}{N_a N_o}\operatorname{diag}(\mathbf{X}^M - \hat{\mathbf{X}}), \tag{99}$$

where $\operatorname{diag}(\mathbf{X}^M - \hat{\mathbf{X}})$ is a diagonal matrix with vector entry $\operatorname{vec}\left(\mathbf{X}^M - \hat{\mathbf{X}}\right)$ on the diagonal.

Consider the term $\frac{\partial \mathbf{X}^0}{\partial \mathbf{W}}$

$$\frac{\partial \mathbf{X}^0}{\partial \mathbf{W}} = \mathbf{X}_{\text{in}} \otimes \mathbf{I}. \tag{100}$$

Therefore, combining Equation (98), (99) and (100) yields

$$\frac{\partial \mathcal{L}}{\partial \mathbf{W}} = \frac{\partial \mathcal{L}}{\partial \mathbf{X}^M} \left[ \mathbf{I} + \Delta t \left( \mathbf{E}_0 + \sum_{i=1}^{M} \mathbf{E}_i \right) + \mathcal{O}(\Delta t^2) \right] \frac{\partial \mathbf{X}^0}{\partial \mathbf{W}}, \tag{101}$$

which reformulates the gradient calculation into a recursive sum. This implies that the gradients will not vanish exponentially, but the gradients may still become very small. □

## B    OTHER PROOFS

We provide additional Lemmas to provide deeper insight into the theoretical properties of our BIMP model.

**Lemma B.1** (Expressive capacity of BIMP). *BIMP can model more diverse node feature representations than approaches whose dynamics are equivalent to linear opinion dynamics.*

*Proof.* Nonlinear systems can exhibit more complex behavior, such as multiple equilibria and bifurcations, than linear systems. Also, many continuous-depth GNNs Chamberlain et al. (2021a); Thorpe et al. (2022); Choi et al. (2023); Nguyen et al. (2024) lack a feature mixing mechanism. In contrast, BIMP introduces the option graph $\mathcal{G}^{\text{o}}$ to enable feature mixing across dimensions, modeling more complex information exchange.

To highlight the contributions of the intrinsic nonlinearity and the option graph $\mathcal{G}^{\text{o}}$, we compare BIMP with linear opinion dynamics as a representative baseline: linear opinion dynamics is a first-order approximation of BIMP without correlated options $\mathbf{A}^{\text{o}} = \mathbf{0}$.

When attention parameter $u = 1$, input parameter $\mathbf{B} = \mathbf{0}$, and uncorrelated options $\mathbf{A}^{\text{o}} = \mathbf{0}$, the BIMP model has dynamics of the form

$$\dot{\mathbf{X}} = -d\mathbf{X} + \tanh\left(\alpha \mathbf{X} + \mathbf{A}^{\text{a}}\mathbf{X}\right). \tag{102}$$

When $\mathbf{X} = \mathbf{0}$, $\dot{\mathbf{X}} = \mathbf{0}$, so $\mathbf{X} = \mathbf{0}$ is an equilibrium of the system. The first-order approximation of our model dynamics about this equilibrium is given by

$$\dot{\mathbf{X}} = (\mathbf{A}^{\text{a}} - c\mathbf{I})\mathbf{X}, \quad c = d - \alpha. \tag{103}$$

When $c = 1$, this equation reduces to,

$$\dot{\mathbf{X}} = (\mathbf{A}^{\text{a}} - \mathbf{I})\mathbf{X}, \tag{104}$$

which is of the same form as linear opinion dynamics in Equation (6). Since linear opinion dynamics has the same form as the first-order approximation of BIMP, we say BIMP has greater expressive capacity.

□

**Lemma B.2** (Expressive capacity of BIMP can degrade). *The BIMP model reduces to a linear system when the attention parameter $u$ is either very small or very large.*

*Proof.* The degeneration to a linear model occurs under two settings: (1) when $u$ is very small and the nonlinear term evaluates to 0; (2) or when $u$ is very large such that the hyperbolic tangent saturates, and therefore the nonlinear term evaluates to $\pm 1$. □

To avoid both degenerate cases, we set the attention parameter $u$ at the bifurcation point. Beyond the reasoning provided in Section 5.2, this lemma offers an additional perspective that placing $u$ at the bifurcation point ensures that BIMP operates within the nonlinear regime of Equation (11), thereby preserving its expressive capacity.

**Lemma B.3** (Reduced order representation of BIMP). *When the input parameter $\mathbf{B}$ is equal to zero, the dynamics of BIMP can be approximated by the dynamics of the reduced one-dimensional dynamical equation.*

$$\dot{y} = -d\,y + \tanh\left[u(\alpha + 3)y\right], \tag{105}$$

*where $y = \langle \mathbf{x}, \mathbf{w}_{\max} \rangle \in \mathbb{R}$, and $\mathbf{w}_{\max}$ is the left dominant eigenvector of $\tilde{\mathbf{A}}$.*

*Proof.* Leveraging the Lyapunov-Schimit reduction, the BIMP dynamics can be projected onto a one-dimensional critical subspace Leonard et al. (2024). The BIMP dynamics in Equation (12) can be vectorized following Lemma 5.2 as

$$\dot{\mathbf{x}} = -d\mathbf{x} + \tanh\left[u\left((\alpha - 1)\mathbf{x} + \tilde{\mathbf{A}}\mathbf{x}\right)\right] + \mathbf{b}, \tag{106}$$

where $\mathbf{x} = \text{vec}(\mathbf{X})$, and $\mathbf{b} = \text{vec}(\mathbf{B})$. Defining the right eigenvector matrix $\mathbf{T} = [\mathbf{v}_1, \mathbf{v}_2, \ldots, \mathbf{v}_{N_a}]$ and the left eigenvector matrix $\mathbf{T}^{-1} = [\mathbf{w}_1, \mathbf{w}_2, \ldots, \mathbf{w}_{N_a}]^\top$ of $\tilde{\mathbf{A}}$, Equation (106) can be expressed in the new coordinates $\mathbf{x} = \mathbf{T}\mathbf{y}$ as

$$\mathbf{T}\dot{\mathbf{y}} = -d\mathbf{T}\dot{\mathbf{y}} + \tanh\left[u\left((\alpha - 1)\mathbf{T}\mathbf{y} + \tilde{\mathbf{A}}\mathbf{T}\mathbf{y}\right)\right]. \tag{107}$$

Multiplying $\mathbf{T}^{-1}$ on both sides gives

$$\dot{\mathbf{y}} = -d\dot{\mathbf{y}} + \mathbf{T}^{-1}\tanh\left[u\left((\alpha - 1)\mathbf{T}\mathbf{y} + \tilde{\mathbf{A}}\mathbf{T}\mathbf{y}\right)\right]. \tag{108}$$

Consider that $c\tanh(x) \approx \tanh(cx)$ for small $|x|$, Equation (108) can be approximated by

$$\dot{\mathbf{y}} = -d\dot{\mathbf{y}} + \tanh\left[u\left((\alpha - 1)\mathbf{y} + \mathbf{T}^{-1}\tilde{\mathbf{A}}\mathbf{T}\mathbf{y}\right)\right]. \tag{109}$$

Defining $\mathbf{\Lambda}$ as the diagonal matrix of eigenvalues of $\tilde{\mathbf{A}}$, Equation (109) can be further simplified by decomposing $\tilde{\mathbf{A}} = \mathbf{T}\mathbf{\Lambda}\mathbf{T}^{-1}$

$$\dot{\mathbf{y}} = -d\dot{\mathbf{y}} + \tanh\left[u\left((\alpha - 1)\mathbf{y} + \mathbf{\Lambda}\mathbf{y}\right)\right]. \tag{110}$$

Equation (110) approximates the dynamics of Equation (106) around $\mathbf{x} = \mathbf{0}$. By observing that $\mathbf{x} = y_1\mathbf{v}_1 + y_2\mathbf{v}_2 + \ldots + y_{N_a}\mathbf{v}_{N_a}$, we can further restrict the dynamics of BIMP to the critical subspace $\text{Ker}(J) = \mathbf{v}_{\max} = \mathbf{v}_1$ through setting $y_2 = y_3 = \ldots = y_{N_a} = 0$. As such, Equation (110) simplifies into

$$\dot{y}_1 = -dy_1 + \tanh\left[u((\alpha - 1)y_1 + \lambda_1 y_1)\right]. \tag{111}$$

Substituting $\lambda_1 = \lambda_{\max}^{\tilde{a}} = 4$ from Proposition 5.3 and simplifying $y_1$ as $y$ gives

$$\dot{y} = -d\dot{y} + \tanh\left[u(\alpha + 3y)\right], \tag{112}$$

which we define as the one-dimensional critical subspace for our model. The remaining eigenvectors $\mathbf{v}_i$ make up the regular subspace as their eigenvalues are smaller than 0. Systems on the regular subspace vanishes quickly and does not contribute to the long-term behavior (i.e, convergence to equilibrium). It is therefore sufficient to focus on the critical subspace to understand the dynamics of the equilibrium as the regular subspace decays quickly. $\square$

**Lemma B.4** (Formation of consensus in BIMP). *BIMP exhibits oversmoothing when the input parameter $\mathbf{B}$ is equal to zero.*

*Proof.* For $\mathbf{x}$ in the neighborhood of the equilibrium $\mathbf{x} = \mathbf{0}$, the Equation (105) in Lemma B.3 is isomorphic to

$$\dot{y} = (u(\alpha + 3) - d)y - u(\alpha + 3)y^3. \tag{113}$$

At the bifurcation point $u = \frac{d}{\alpha + 3}$, Equation (113) has unique equilibrium $y = 0$.

This corresponds to an equilibrium solution of $\mathbf{x} = \mathbf{0}$ in the original system (Equation (36)) which means that $\mathbf{X} = \mathbf{0}$ and all agents form neutral opinions for all options. Since the opinions of all agents have converged, the system has reached consensus (i.e., exhibits oversmoothing). $\square$

This lemma indicates that BIMP requires an appropriately chosen input term $\mathbf{B}$ to avoid converging to consensus, as discussed in Theorem 5.6.

## C  DATASETS

### C.1  HOMOPHILIC DATASETS

Table 2 in Section 6.2 performs semi-supervised node classification task on the Cora McCallum et al. (2000), Citeseer Sen et al. (2008), Pubmed Namata et al. (2012), CoauthorCS Shchur et al. (2018) and Amazon Computers and Photo McAuley et al. (2015) homophilic datasets. Recent continuous-depth models such as GRAND-$\ell$ and KuramotoGNN make use of the largest connected component (LCC) of the datasets, which discards smaller subgraphs. However, Wu et al. (2022) theoretically analyzes that, for dense graphs, the number of GNN layers required to trigger oversmoothing decreases as the graph size shrinks. This creates a trade-off between avoiding smaller components to oversmooth and building deep enough models for expressivity. Instead, since our BIMP is designed to mitigate oversmoothing across all parts of graphs, we retain the full graph and show that BIMP maintains stable performance.

**Cora.** The Cora dataset contains a citation graph where 2708 computer science publications are connected by 5278 citation edges. Each publication has an 1433-dimensional bag-of-words vector derived from a paper keyword dictionary. Publications are classified into one of 7 classes corresponding to their primary research area.

**Citeseer.** The Citeseer dataset contains a citation graph where 3312 computer science publications are connected by 4552 citation edges. Each publication has a 3703-dimensional bag-of-words vector derived from a paper keyword dictionary. Publications are classified into one of 6 classes corresponding to their primary research area.

**Pubmed.** The Pubmed dataset contains a citation graph where 19717 biomedical publications are connected by 44324 citation edges. Each publication is represented by a 500-dimensional TF/IDF weighted word vector derived from a paper keyword dictionary. Publications are classified into one of 3 classes corresponding to their primary research area.

**CoauthorCS.** The CoauthorCS dataset is one segment of the Coauthor Graph datasets that contains a co-authorship graph that consist of 18333 authors and connected by 81894 co-authorship edges. Each author is represented by a 6805-dimensional bag-of-words feature vector derived from their paper keywords. Authors are classified into one of 15 classes corresponding to their primary research area.

**Amazon Computers.** The Amazon Computers dataset, denoted as Computers in our paper, contains a co-purchase graph where 13381 computer products are connected by 81894 edges. The edges indicate that two products are frequently bought. Each product is represented by a 767-dimensional bag-of-words feature vector derived from their product reviews. Products are classified into one of 10 classes corresponding to their product categories.

**Amazon Photo.** The Amazon Photo dataset, denoted as Photo in our paper, contains a co-purchase graph where 7487 photo products are connected by 119043 edges. The edges indicate that two products are frequently bought. Each product is represented by a 745-dimensional bag-of-words feature vector derived from their product reviews. Products are classified into one of 8 classes corresponding to their product categories.

### C.2  HETEROPHILIC DATASETS

Table 6 in Section 6.2 performs semi-supervised node classification task on the Texas, Wisconsin, and Cornell heterophilic datasets from the CMU WebKB Craven et al. (1998) project.

**Texas.** The Texas dataset contains a webpage graph where 183 web pages are connected by 325 hyperlink edges. Each webpage has a 1703-dimensional bag-of-words vector derived from the contents of the webpage. Webpages are classified into one of 5 classes corresponding to their primary content.

**Wisconsin.** The Wisconsin dataset contains a webpage graph where 251 web pages are connected by 512 hyperlink edges. Each webpage has a 1703-dimensional bag-of-words vector derived from the contents of the webpage. Webpages are classified into one of 5 classes corresponding to their primary content.

**Cornell.** The Cornell dataset contains a webpage graph where 183 web pages are connected by 298 hyperlink edges. Each webpage has a 1703-dimensional bag-of-words vector derived from the contents of the webpage. Webpages are classified into one of 5 classes corresponding to their primary content.

**Squirrel.** The Squirrel dataset contains a Wikipedia page-page network on squirrels where 5201 pages are connected by 217073 edges. Each webpage has a 2089-dimensional bag-of-words vector derived from the contents of the webpage. Nodes are classified into one of 5 classes in term of the number of the average monthly traffic of the web page.

**Chameleon.** The Chameleon dataset contains a Wikipedia page-page network on chameleon where 2277 pages are connected by 36101 edges. Each webpage has a 2325-dimensional bag-of-words vector derived from the contents of the webpage. Nodes are classified into one of 5 classes in term of the number of the average monthly traffic of the web page.

**Actor.** The Actor dataset contains the actor-only induced subgraph of the film-director-actor-writer network where 7600 actors are connected by 30019 co-occurrence on Wikipedia. Each webpage has a 932-dimensional bag-of-words vector derived from the contents of the webpage. Nodes are classified into one of 5 classes in term of words of actor's Wikipedia page.

### C.3 LARGE GRAPHS

Table 7 in Appendix D.2.3 performs semi-supervised node classification task on the ogbn-arXiv Hu et al. (2020) dataset. The ogbn-arXiv dataset consists of a single graph with 169,343 nodes and 1,166,243 edges where each node represents an arxiv paper, and edges represent citation relationships. We train each model in a semi-supervised way, and compute the training loss over 90,941 of the 169,343 nodes. We use 29,799 of the remaining nodes for validation, and the final 48,603 nodes for testing.

# D EXPERIMENT DETAILS

## D.1 PERFORMANCE AT LARGE DEPTHS

### D.1.1 CLASSIFICATION ACCURACY

Experiment 6.1: Classification accuracy evaluates the classification performance of BIMP and continuous-depth baselines at different depths of $T = \{1, 2, 4, 8, 16, 32, 64, 128\}$ with 100 splits and 10 random seeds. We use the classification accuracy as a measure of the robustness to deep layers of BIMP and baseline methods.

Figure 3 show the comparison of classification accuracy of BIMP and select continuous baseline methods. Figure 5 show the comparison of classification accuracy of BIMP and additional baselines and oversmoothing mitigation techniques, including pairnorm (Zhao & Akoglu, 2019) (denoted -pairnorm) and differentiable group normalization (Zhou et al., 2020) (denoted -group). Specially, in differentiable group normalization, we have the skip-connection as suggested in their paper.

Since adaptive step-size methods like Dormand–Prince (Dopri5) can result in inconsistent numbers of integration steps, we implement the Euler method with fixed step size $\Delta t = 1$ for fair comparison. Notably, BIMP outperforms GRAND++-$\ell$ at significant depths, even though GRAND++-$\ell$ only supports Dormand–Prince (Dopri5).

Some baselines incorporate an additional learnable weight to scale the differential equation. For instance, in GRAND-$\ell$, the implementation was modified as $\dot{\mathbf{x}} = \alpha[(A - I)\mathbf{x} + \beta\mathbf{x}(0)]$, where $\alpha$ acts as a time-scaling factor. To eliminate its influence and ensure consistency, we set $\alpha = 1$ across all methods.

For each method, we use the fine-tuned parameters provided by each baseline and fix the set of hyperparameters across all depths. For all experiments, we run 100 train/valid/test splits for each dataset with 10 random seeds for each split.

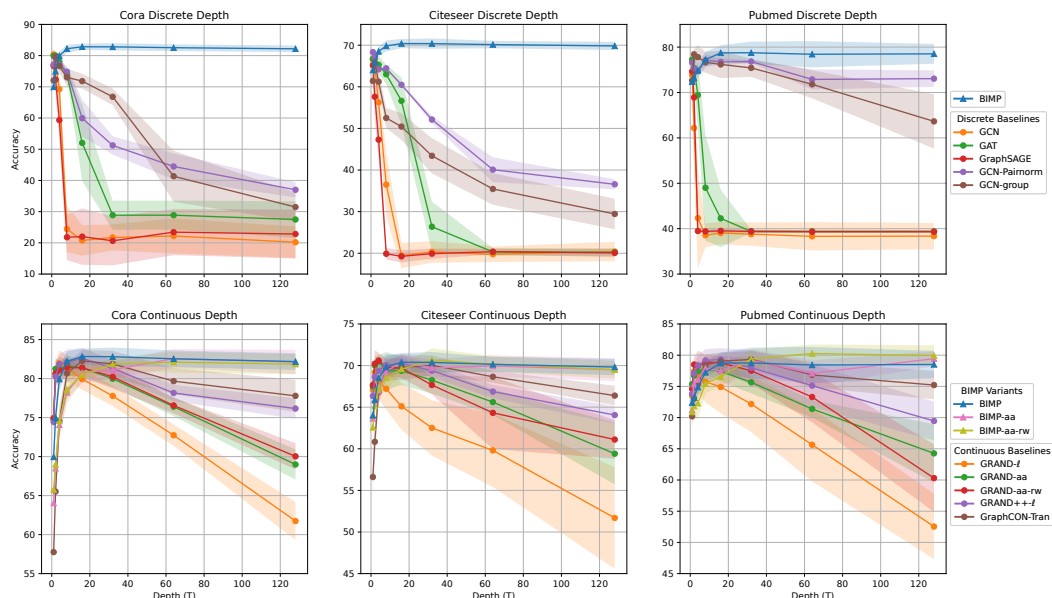

Figure 5: **Classification accuracy.** BIMP is designed to learn node representations that resist oversmoothing even for very large depths. We compare the classification accuracy of BIMP to baseline models for architectures with $1, 2, 4, 8, 16, 32, 64$ and $128$ timesteps. Our BIMP model and its variants are stable out to 128 timesteps, while baseline performance deteriorates after 32 timesteps.

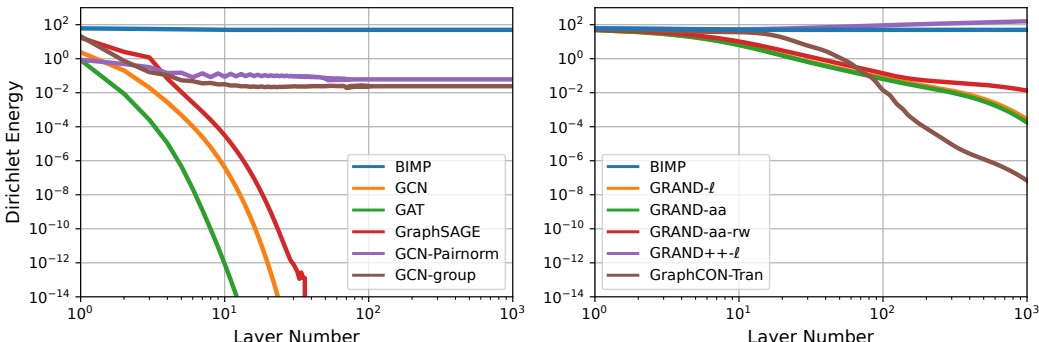

Figure 6: **Dirichlet energy.** BIMP is designed to learn node representations that resist oversmoothing even for very large depths. We compare the Dirichlet energy of node features over a range of network depths. The Dirichlet energy of BIMP remains stable even at very deep layers, while the energy of baseline modes does not.

### D.1.2 DIRICHLET ENERGY

Experiment 6.1: Dirichlet energy illustrates the dynamics of the Dirichlet energy in BIMP and baseline methods, which indicates the similarity between the learned features. We randomly generate an undirected graph with 10 nodes each with 2-dimensional features sampled from $\mathcal{U}[0, 1]$. We randomly initialize the models with the same seed and the node features are propagated forward through 1000 layers.

Figure 3 show the comparison of Dirichlet energy of BIMP and select continuous baseline methods. Figure 6 show the comparison of Dirichlet energy of BIMP and additional baselines and oversmoothing mitigation techniques.

## D.2 CLASSIFICATION ACCURACY

### D.2.1 CLASSIFICATION ACCURACY ON HOMOPHILIC DATASET

We demonstrate improved classification performance compared to GRAND-$\ell$, GRAND++-$\ell$, KuramotoGNN, GraphCON, GAT, GCN and GraphSAGE. Additionally, we consider oversmoothing mitigation techniques of pairnorm Zhao & Akoglu (2019) (denoted -pairnorm) and differentiable group normalization Zhou et al. (2020) (denoted -group) and training methods of adaptive attention (denoted -aa) and rewiring with graph diffusion convolution Atwood & Towsley (2016) (denoted -rw). Classification performance on random splits is reported in Table 4, and performance on standard splits is reported in Table 5, where BIMP constantly outperforms baseline methods.

Table 4: **Classification accuracy on homophilic datasets.** Classification accuracies on the Cora, Citeseer, Pubmed, CoauthorCS, Computers, and Photo datasets are reported, where BIMP outperforms competitive baselines. Results are averaged over 20 random initializations and 100 random train–validation–test splits.

| Dataset | Cora | Citeseer | Pubmed | CoauthorCS | Computers | Photo |
|---|---|---|---|---|---|---|
| **BIMP** | **83.19±1.13** | **71.09±1.40** | 80.16±2.03 | **92.48±0.26** | 84.73±0.61 | 92.90±0.44 |
| **BIMP-aa** | 82.96±1.31 | 70.43±0.80 | **80.35±0.99** | 91.82±0.37 | 84.72±0.45 | 92.27±0.36 |
| **BIMP-aa-rw** | 82.59±1.06 | 70.51±1.37 | 78.56±1.12 | 91.97±0.37 | **84.76±0.23** | **92.92±0.19** |
| GRAND-$\ell$ | 82.20±1.45 | 69.89±1.48 | 78.19±1.88 | 90.23±0.91 | 82.93±0.56 | 91.93±0.39 |
| GRAND-aa | 82.59±0.28 | 70.21±1.21 | 78.39±1.95 | 91.44±0.42 | 83.09±1.71 | 92.50±0.53 |
| GRAND-aa-rw | 82.86±1.47 | 70.95±1.13 | 78.56±1.13 | 91.52±0.31 | 83.47±0.51 | 92.64±0.24 |
| GRAND++-$\ell$ | 82.83±1.31 | 70.26±1.46 | 78.89±1.96 | 90.10±0.78 | 82.79±0.54 | 91.51±0.41 |
| GRAND++-aa | 80.14±0.93 | 69.94±1.45 | 78.50±1.28 | 85.65±1.30 | 84.00±0.47 | 91.86±0.52 |
| GRAND++-aa-rw | 81.91±1.39 | 69.41±0.95 | 79.44±1.06 | 86.23±0.80 | 83.35±0.63 | 92.50±0.22 |
| KuramotoGNN | 81.16±1.61 | 70.40±1.02 | 78.69±1.91 | 91.05±0.56 | 80.06±1.60 | 92.77±0.42 |
| GraphCON-Tran | 82.80±1.34 | 69.60±1.16 | 78.85±1.53 | 90.30±0.74 | 82.76±0.58 | 91.78±0.50 |
| GAT | 79.76±1.50 | 67.70±1.63 | 76.88±2.08 | 89.51±0.54 | 81.73±1.89 | 89.12±1.60 |
| GCN | 80.76±2.04 | 67.54±1.98 | 77.04±1.78 | 90.98±0.42 | 82.02±1.87 | 90.37±1.38 |
| GCN-pairnorm | 79.55±1.21 | 66.93±0.94 | 76.14±0.63 | 90.63±0.69 | 81.88±2.73 | 86.93±1.35 |
| GCN-group | 80.48±1.40 | 66.99±1.97 | 77.53±0.97 | 90.97±0.54 | 81.97±0.75 | 89.84±0.71 |
| GraphSAGE | 79.37±1.70 | 67.31±1.63 | 75.52±2.19 | 90.62±0.42 | 76.42±7.60 | 88.71±2.68 |

Table 5: **Classification accuracy on Planetoid datasets.** We report the classification accuracies on the Cora, Citeseer and Pubmed, using 20 different initializations on the Planetoid train-val-test splits, where BIMP outperforms competitive baselines.

| Dataset | Cora | Citeseer | Pubmed |
|---|---|---|---|
| **BIMP** | 83.45±0.61 | **72.52±0.28** | **80.18±0.63** |
| **BIMP-aa** | 82.81±0.62 | 71.73±1.18 | 80.53±0.82 |
| **BIMP-aa-rw** | 82.23±0.72 | 72.21±0.77 | 79.52±0.28 |
| GRAND-$\ell$ | **83.60±0.56** | 71.29±0.74 | 79.76±0.28 |
| GRAND++-$\ell$ | 83.31±0.74 | 71.84±0.57 | 79.23±0.69 |
| KuramotoGNN | 83.26±1.13 | 71.31±0.62 | 79.79±0.49 |
| GraphCON-Tran | 82.42±0.60 | 71.56±1.09 | 79.92±0.61 |
| GAT | 80.49±0.74 | 65.55±0.76 | 77.70±0.35 |
| GCN | 81.89±0.63 | 66.26±0.56 | 77.64±0.50 |
| GCN-pairnorm | 79.85±1.33 | 66.25±1.54 | 76.28±0.36 |
| GCN-group | 81.13±0.04 | 67.60±1.02 | 77.87±0.49 |

### D.2.2 CLASSIFICATION ACCURACY ON HETEROPHILIC DATASET

We demonstrate overall improved classification performance compared to GRAND-$\ell$, GRAND++-$\ell$, KuramotoGNN, GraphCON, GAT, GCN, and GraphSAGE, as well as superior performance on larger datasets compared to heterophily-specific methods such as ACM-GNN (Luan et al., 2022) and GloGNN (Li et al., 2022). Additionally, we consider oversmoothing mitigation techniques of pairnorm (denoted -pairnorm) and differentiable group normalization (denoted -group) and training methods of adaptive attention (denoted -aa) and rewiring with graph diffusion convolution (denoted -rw). Classification performance is reported in Table 6.

Table 6: **Classification accuracy on heterophilic datasets.** Classification accuracies on theree small datasets, Texas, Wisconsin, and Cornell, and three larger datasets, Actor, Squirrel, Chameleon, are reported, where BIMP outperforms competitive baselines, especially for the larger datasets. Results are averaged over 100 random initializations and 10 standard splits. We adopt some baseline results reported in the GloGNN paper, marked with $^\dagger$.

| Dataset | Cornell | Texas | Wisconsin | Actor | Squirrel | Chameleon |
|---|---|---|---|---|---|---|
| *Homophily level* | 0.30 | 0.11 | 0.21 | 0.22 | 0.22 | 0.23 |
| **BIMP** | 77.13±3.38 | 82.16±4.06 | 86.57±4.33 | 37.46±1.24 | **58.22±1.16** | 69.72±1.81 |
| **BIMP-aa** | 76.95±4.71 | 82.25±6.49 | 86.27±4.36 | **37.52±1.08** | 58.17±1.12 | **69.79±1.77** |
| **BIMP-aa-rw** | 77.46±4.80 | 82.43±6.76 | 86.22±4.34 | 36.78±0.65 | 55.18±1.67 | 69.10±1.98 |
| GRAND-$\ell$ | 70.00±6.22 | 74.59±5.43 | 82.75±3.90 | 36.68±1.25 | 41.11±1.70 | 55.61±1.97 |
| GRAND++-$\ell$ | 70.30±8.50 | 76.14±5.77 | 83.09±2.83 | 34.28±1.98 | 34.68±1.60 | 50.44±1.77 |
| KuramotoGNN | 76.02±2.77 | 81.81±4.36 | 85.09±4.42 | 35.67±1.28 | 36.22±1.76 | 50.63±2.00 |
| GraphCON-GCN | 74.05±3.24 | 80.54±4.49 | 84.79±2.51 | 35.69±1.04 | 31.53±1.46 | 41.18±1.53 |
| GloGNN | 83.35±4.42 | 81.30±6.28 | 85.57±4.36 | 37.26±1.57 | 57.48±1.63 | 69.68±2.55 |
| ACM-GNN | **91.95±4.32** | **90.41±4.16** | **92.94±3.99** | 37.32±1.37 | 56.83±1.99 | 67.69±2.21 |
| GAT | 42.16±7.07 | 57.84±5.82 | 49.61±4.21 | 27.44±0.89$^\dagger$ | 40.72±1.55$^\dagger$ | 60.26±2.50$^\dagger$ |
| GCN | 41.35±4.69 | 57.03±5.98 | 48.43±5.75 | 27.32±1.10$^\dagger$ | 53.43±2.01$^\dagger$ | 64.82±2.24$^\dagger$ |

### D.2.3 EXPERIMENT ON LARGE GRAPH

We demonstrate improved classification performance compared to GRAND-$\ell$, GRAND++-$\ell$, KuramotoGNN, GraphCON, GCN, GAT and GraphSAGE on ogbn-arXiv dataset, where BIMP outperforms all continuous-depth baseline methods.

Table 7: **Classification accuracy on ogbn-arXiv dataset.** Our BIMP model outperforms GRAND-$\ell$, GRAND++-$\ell$, KuramotoGNN, GraphCON, GCN, GAT and GraphSAGE on the ogbn-arXiv dataset, using 20 random initialization on the standard split.

| Dataset | ogbn-arXiv | number of parameters |
|---|---|---|
| **BIMP** | **71.04±0.94** | 128,159 |
| GRAND-$\ell$ | 70.19±0.43 | 98,964 |
| GRAND++-$\ell$ | 67.61±0.34 | 320,791 |
| KuramotoGNN | 66.96±0.25 | 160,719 |
| GraphCON-Tran | 67.13±0.41 | 99,290 |
| GCN | 61.66±0.32 | 200,967 |
| GAT | 69.86±0.59 | 435,733 |
| GraphSAGE | 66.51±0.26 | 401,671 |

### D.2.4 LONG-RANG GRAPHS EXPERIMENT.

We demonstrate improved prediction performance compared to GCN, GIN, Transformer+LapPE, SAN+LapPE, GRAND-$\ell$, GraphCON and ADGN on Peptides-func and Peptides-struc datasets, where BIMP outperforms all continuous-depth baseline methods and on par with the SOTA results (Table 3).

Following the experiment design in Gravina et al. (2025), we evaluated our BIMP model with 3 different seeds, and baseline results are taken directly from Gravina et al. (2025).

According to the result, Graphormers with Laplacian positional encoding (denoted as LapPE) achieve slightly better performance than our BIMP. However, LapPE requires a preprocessing step. For the Peptides-struct dataset, this preprocessing amounts to 1m 14s on an NVIDIA A100 with four AMD Milan 74133 CPU cores Dwivedi et al. (2022). In addition, BIMP requires only ∼111K parameters, which is substantially smaller than Graphormer's ∼500K, highlighting the efficiency of our architecture.

### D.2.5 HOMOPHILIC DATASET HYPERPARAMETERS

We search the hyperparameters using Ray Tune Liaw et al. (2018) with 1000 random trials for each dataset and final values are shown in Table 8. Experiments were run with 100 random splits and each split trained on 20 seeds.

Table 8: **Hyperparameter for homophilic dataset.** The hyperparameters for homophilic datasets in Section 6.2 is reported.

| Dataset | Cora | Citeseer | Pubmed | CoauthorCS | Computers | Photo |
|---|---|---|---|---|---|---|
| Opinion Dim. | 80 | 128 | 128 | 16 | 128 | 64 |
| Epoch | 100 | 250 | 600 | 250 | 100 | 100 |
| Learning Rate | 0.0178 | 0.0034 | 0.0210 | 0.0018 | 0.0035 | 0.0056 |
| Optimizer | AdaMax | AdaMax | AdaMax | RMSProp | Adam | Adam |
| Weight Decay | 0.0078 | 0.1 | 0.0020 | 0.0047 | 0.0077 | 0.0047 |
| Dropout | 0.1353 | 0.3339 | 0.0932 | 0.6858 | 0.0873 | 0.4650 |
| Input Dropout | 0.4172 | 0.5586 | 0.6106 | 0.5275 | 0.5973 | 0.4290 |
| Attention Head | 4 | 2 | 1 | 4 | 4 | 4 |
| Attention Dim. | 16 | 8 | 16 | 8 | 64 | 64 |
| Attention Type | Scaled Dot | Exp. Kernel | Cosine Sim. | Scaled Dot | Scaled Dot | Pearson |
| NODE Adjoint | False | False | True | True | True | True |
| Adjoint Method | n/a | n/a | Euler | dopri5 | dopri5 | dopri5 |
| Adjoint Step Size | n/a | n/a | 1 | 1 | 1 | 1 |
| Integral Method | dopri5 | dopri5 | dopri5 | dopri5 | dopri5 | dopri5 |
| Linear Encoder | True | True | False | True | True | False |
| Linear Decoder | True | True | True | True | True | True |
| Step Size | 1 | 1 | 1 | 1 | 1 | 1 |
| Time ($T$) | 12.2695 | 6.6067 | 9.7257 | 4.0393 | 3.2490 | 2.0281 |
| Damping ($d$) | 0.8952 | 1.0970 | 0.6908 | 0.1925 | 1.0269 | 1.0230 |
| Self-reinforce ($\alpha$) | 1 | 1 | 1 | 1 | 1 | 1 |

### D.2.6 HETEROPHILIC DATASET HYPERPARAMETERS

We search the hyperparameters using Ray Tune with 200 random trials for each dataset and final values are shown in Table 9. Experiments were run with 10 standardized splits and each split trained on 100 seeds.

## E ADDITIONAL EXPERIMENTS

### E.1 MULTI-AGENT TRAJECTORY EXTRAPOLATION

In the motivating experiment (Figure 1), we observe that GCN-based GraphODE (Poli et al., 2019) tends to collapse trajectories, thereby degrading predictive accuracy and underscoring the critical role of oversmoothing. Here, we provide the detail of the experiment setup, model architecture and additional results of both GraphODE and BIMP.

Table 9: **Hyperparameter for heterophilic dataset.** The hyperparameters for heterophilic datasets in Section 6.2 is reported.

| Dataset | Texas | Wisconsin | Cornell |
|---|---|---|---|
| Opinion Dim. | 256 | 32 | 32 |
| Epoch | 200 | 100 | 100 |
| Learning Rate | 0.0178 | 0.0178 | 0.0218 |
| Optimizer | AdaMax | AdaMax | AdaMax |
| Weight Decay | 0.0078 | 0.0091 | 0.0478 |
| Dropout | 0.6531 | 0.2528 | 0.2030 |
| Input Dropout | 0.0052 | 0.0042 | 0.0417 |
| Attention Head | 8 | 4 | 4 |
| Attention Dim. | 32 | 16 | 16 |
| Attention Type | Scaled Dot | Scaled Dot | Scaled Dot |
| NODE Adjoint | False | False | False |
| Integral Method | dopri5 | dopri5 | dopri5 |
| Linear Encoder | False | False | False |
| Linear Decoder | True | False | False |
| Step Size | 1 | 1 | 1 |
| Time ($T$) | 0.01 | 0.01 | 0.01 |
| Damping ($d$) | 0.0086 | 0.0075 | 0.0195 |
| Self-reinforce ($\alpha$) | 2 | 2 | 1.5 |

### E.1.1 EXPERIMENT SETUP

We evaluate BIMP and the GCN-based GraphODE on extrapolating the dynamics of a synthetic mass–spring system, where 2D particles are randomly connected by springs with connection probability $\frac{1}{2}$. Each edge is binary (0/1), indicating the presence or absence of a spring. The node features are the position $\mathbf{x}_i$ and velocity $\mathbf{v}_i$ of agent $i$. The system dynamics are governed by

$$\dot{\mathbf{x}}_i = \mathbf{v}_i, \qquad \dot{\mathbf{v}}_i = - \sum_{j \in \mathcal{N}(i)} k_{ij}(\mathbf{x}_i - \mathbf{x}_j) \tag{114}$$

where $k_{ij}$ is the interaction strength between agent $i$ and $j$.

Starting from the initial states, the models are required to recursively extrapolate the next 19 steps. Both BIMP and GraphODE are trained by minimizing the mean squared error between the predicted and ground-truth particle states across these 19 steps. For evaluation, we deploy the trained models on the test set to recursively predict the next 19 steps, and plot the predicted trajectories separately.

### E.1.2 MODEL ARCHITECTURE

Inspired by Poli et al. (2019); Huang et al. (2020), we design both GraphODE and BIMP using an encoder–processor–decoder architecture. The encoder $\phi$ maps the initial state $\mathbf{x}_i(t_0)$ into the latent space as $\mathbf{z}_i(t_0)$, after which an ODE function $g$ predicts latent trajectories starting from the encoded initial states. Finally, a decoder $\psi$ reconstructs the predicted dynamics $\mathbf{x}_i(t)$ at any timestamp $t$. The architecture can be summarized as

$$\mathbf{z}_i(t_0) = \phi(\mathbf{x}_i(t_0)), \tag{115}$$

$$\mathbf{z}_i(t) = \mathbf{z}_i(t_0) + \int_{t_0}^{t} g\big(\mathbf{z}_i(t), \mathcal{G}\big) dt, \tag{116}$$

$$\mathbf{x}_i(t) = \psi(\mathbf{z}_i(t)). \tag{117}$$

where $\mathcal{G}$ is the graph. In GraphODE, $g$ is implemented as a two-layer GCN, while in BIMP, $g$ is instantiated as our nonlinear opinion dynamics model (Equation 11). We employ the Euler method for numerical integration. Notably, since $g$ is recursively applied during latent trajectory prediction, this process is equivalent to stacking many layers, which makes the model susceptible to oversmoothing in the latent space.

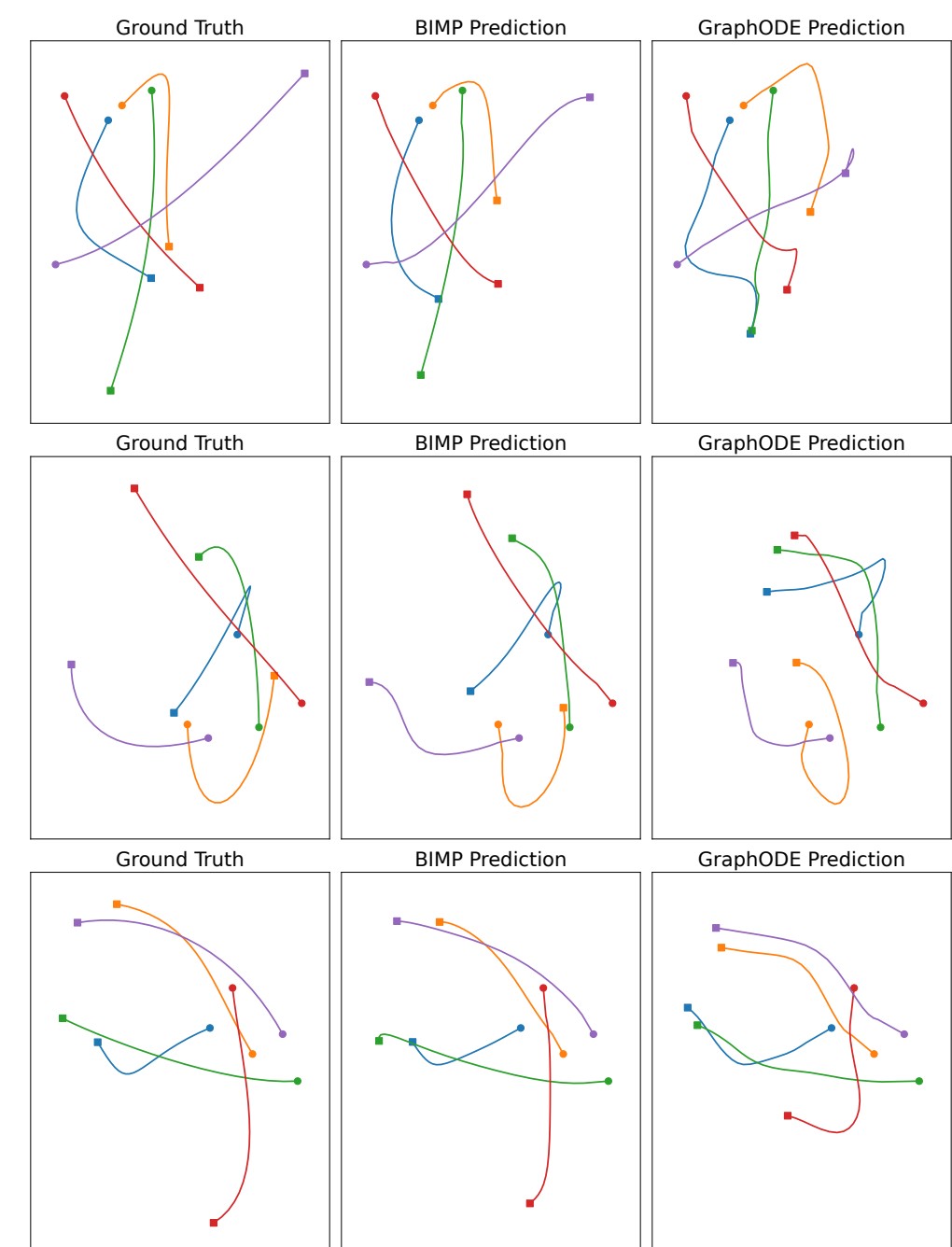

Figure 7: **Mass–spring trajectory extrapolation.** Each row presents trajectories predicted by BIMP (**middle**) and GraphODE (**right**), with the ground-truth trajectories shown in the left column for reference. We observe that, over time, predictions of GraphODE collapse to the same position, deviating from the ground truth trajectories.

### E.1.3    ADDITIONAL RESULTS

We present three additional prediction examples in Figure 7. Over time, GraphODE predictions collapse toward the same region, deviating dramatically from the ground-truth trajectories. In contrast, BIMP remains robust to oversmoothing, maintains well-separated trajectories, and achieves superior prediction accuracy. Furthermore, we quantify the predict performance by MSE and mean average

percentage error (MAPE) in Table 10, computing the error at each prediction time step. BIMP consistently outperforms GraphODE across different steps.

Table 10: **Accuracy across predicted trajectories.** We report the MSE and mean absolute percentage error (MAPE) on the mass-spring trajectory predicting task, averaged over 1000 synthetic trajectories. BIMP consistently outperforms GraphODE across different predict steps.

| Steps | 2 | 4 | 8 | 14 | 20 |
|---|---|---|---|---|---|
| **MSE** | | | | | |
| BIMP (ours) | 0.0015±0.0010 | 0.0019±0.0009 | 0.0083±0.0045 | 0.0235±0.0152 | 0.0331±0.0242 |
| GraphODE | 0.0081±0.0043 | 0.0082±0.0049 | 0.0217±0.0123 | 0.0873±0.0482 | 0.2379±0.1128 |
| **MAPE** | | | | | |
| BIMP (ours) | 27.51%±4.98% | 38.04%±12.81% | 57.54%±26.15% | 94.22%±46.64% | 144.60%±51.52% |
| GraphODE | 70.01%±16.8% | 129.55%±69.01% | 123.84%±78.50% | 356.08%±107.20% | 610.97%±260.51% |

### E.2 COMPUTATIONAL COMPLEXITY

BIMP's space complexity is higher than other baselines as its increase in expressive capacity comes from the introducing of an additional option adjacency matrix $\mathbf{A}^o$. Specially, we compare the space complexity of BIMP with that of GRAND-$\ell$. The complexity of BIMP is $\mathcal{O}(mN_o + N_o^2)$, where $m$ is the number of edges, $N_a$ is the number of nodes, and $N_o$ is the number of options. The space complexity of GRAND-$\ell$ is $\mathcal{O}(mN_o)$. Given that the number of options is generally smaller than the number of agents, $N_o$ remains relatively small compared to the number of edge $m$, resulting acceptable computational overhead.

Additionally, BIMP introduces a nonlinearity through the saturation function $\tanh(\cdot)$, which increases computational cost. However, considering this function operates element-wise, it is more efficient than other nonlinear dynamical processes, such as KuramotoGNN. To illustrate this difference, we compare the average run time of our model against baseline models in Table 11. We note that BIMP has comparable run time performance to linear baselines models such as GRAND++-$\ell$.

We record the running time for BIMP and other 4 popular continuous-depth GNNs, GRAND-$\ell$, GRAND++-$\ell$, GraphCON-Tran and KuramotoGNN on Cora and Citeseer dataset. We train each model 100 times with with a fixed number of epoch (100 for Cora and 250 for Citeseer) using fine tuned hyperparameters. The average training time listed in Table 11 demonstrates, in contrast to other nonlinear methods like KuramotoGNN, BIMP maintains a training time comparable to other linear continuous-depth baselines.

To compare against competitive non-ODE baselines, we report the running time for BIMP and GCN-residual (Chi et al., 2021), GATv2 (Brody et al., 2022) and GOAT (Kong et al., 2023) on the ogbn-arXiv dataset. Each model is trained 10 times for 100 epochs using fine tuned hyperparameters. The average running time and memory usage reported in Table 12 demonstrates, in contrast to other transformer based method like GOAT, BIMP maintains a training time comparable to other non-ODE baselines and requires the least amount of memory.

All experiments reported in the paper was conducted on work stations with an Intel Xeon Gold 5220R 24 core CPU, an Nvidia A6000 GPUs, and 256GB of RAM.

Table 11: **Comparable running time with ODE based methods.** The average running time (in seconds) for each fine-tuned method tested on the Cora dataset for 100 epochs and the Citeseer dataset for 250 epochs. Our BIMP model exhibits a modest increase in running time.

| Dataset | BIMP (ours) | GRAND-$\ell$ | GRAND++-$\ell$ | GraphCON-Tran | KuramotoGNN |
|---|---|---|---|---|---|
| Cora | 14.33 | 11.97 | 14.06 | 12.86 | 201.32 |
| Citeseer | 42.38 | 31.34 | 41.47 | 15.84 | 252.73 |

### E.3 CHOICE OF NONLINEARITY IN NOD MODULE

To understand how the choice of nonlinearity in our Nonlinear Opinion Dynamics (NOD) module impacts performance, we experiment with a suite of alternative nonlinearities (softsign, arctan,

Table 12: **Comparable running time with non-ODE based methods.** The average running time (in seconds) and memory usage for each fine-tuned method tested on the ogbn-arXiv dataset for 100 epochs. Numbers in parentheses indicate the additional time for the positional encoding preprocessing. Our BIMP model exhibits a modest increase in running time and requires the least memory.

| | BIMP (ours) | GCN-residual | GATv2 | GOAT |
|---|---|---|---|---|
| Running time (s) | 114.15 | 37.43 | 77.81 | 3601.13 (75.96) |
| Memory usage (MB) | 1465 | 4628 | 6975 | 3210 |

sigmoid, ReLu and GELU) and linearity (linear). Softsign and arctan satisfy the nonlinearity constraint in the NOD definition (i.e., $S(0) = 0, S'(0) = 1, S''(0) \neq 0$), but sigmoid, ReLu and GELU do not. Specifically, sigmoid does not pass through the origin, ReLu is not differentiable, and GELU does not satisfy $S'(0) = 1$. Linear refers to the BIMP model without any nonlinearity. We find that using nonlinearities that meet the NOD criteria effectively prevent oversmoothing, while the others do not. We report the classification accuracy of our BIMP model with alternative nonlinearities in the NOD module in Table 13.

For better understanding, we also visualize the Dirichlet energy of node features over a range of network depths, given different choices of nonlinearity. The result is shown in Figure 8 (left). We observe the BIMPs without satisfying saturation functions make the Dirichlet energy explode significantly, while the BIMPs with reasonable saturation functions stabilize the Dirichlet energy.

Table 13: **Nonlinear opinion dynamics nonlinearity ablation.** Classification accuracy of our BIMP model on the Cora dataset using various nonlinearities.

| Layer | tanh | softsign | arctan | sigmoid | ReLu | GELU | linear |
|---|---|---|---|---|---|---|---|
| 1 | 69.96±1.45 | 63.25±1.73 | 64.05±1.58 | 60.36±1.39 | 64.17±1.72 | 62.45±1.57 | **77.52±1.44** |
| 2 | 75.00±1.50 | 68.34±2.03 | 72.62±2.40 | 63.20±1.63 | 67.21±2.18 | 67.06±2.25 | **81.92±0.85** |
| 4 | 79.93±1.41 | 72.37±1.50 | 76.91±1.82 | 65.54±1.80 | 73.16±1.44 | 72.40±1.60 | **82.08±1.34** |
| 8 | **82.21±1.26** | 77.05±1.67 | 79.84±1.05 | 63.91±2.13 | 77.76±1.47 | 77.56±1.49 | 81.24±1.48 |
| 16 | **82.83±1.12** | 79.88±1.82 | 81.81±1.54 | 29.55±1.80 | 81.32±0.96 | 81.47±0.63 | 80.45±1.43 |
| 32 | **82.81±1.19** | 81.45±1.48 | 82.48±1.81 | 29.92±1.22 | 82.51±1.51 | 27.99±4.22 | 79.99±1.21 |
| 64 | 82.53±1.07 | 81.14±1.65 | **82.94±0.73** | 30.72±1.02 | 76.56±3.88 | 29.64±1.82 | 76.74±1.86 |
| 128 | **82.18±1.06** | 81.71±1.37 | 81.26±1.93 | 29.37±2.77 | 71.18±7.09 | 26.27±4.89 | 75.44±0.89 |

Table 14: **Inductive bias ablation.** Classification accuracy of our BIMP model on the Cora dataset.

| Layer | BIMP | w/o damping term | w/o input | w/o attention mechanism | w/o right-stochastic |
|---|---|---|---|---|---|
| 1 | 69.96±1.45 | 67.83±1.55 | **74.25±1.25** | 72.21±2.18 | 72.96±0.86 |
| 2 | 75.00±1.50 | 74.00±1.25 | **79.40±1.77** | 78.09±1.17 | 79.17±1.62 |
| 4 | 79.93±1.41 | 78.53±1.46 | **81.74±1.32** | 80.22±1.95 | 81.32±1.49 |
| 8 | **82.21±1.26** | 78.38±0.98 | 81.32±1.07 | 81.31±2.24 | 82.03±0.98 |
| 16 | **82.83±1.12** | 79.62±1.37 | 81.24±1.64 | 81.85±1.20 | 82.71±1.22 |
| 32 | **82.81±1.19** | 79.12±1.15 | 79.94±0.82 | 81.45±1.13 | 82.32±1.32 |
| 64 | **82.53±1.07** | 78.95±1.62 | 77.71±0.82 | 81.28±1.37 | 80.44±1.40 |
| 128 | **82.18±1.06** | 76.60± 2.27 | 74.10±0.92 | 81.04±1.15 | 71.17±5.37 |

### E.4 INDUCTIVE BIAS ABLATION

We ablate the proposed nonlinear opinion dynamics inductive bias by removing the input $\mathbf{B}$, damping term $d$, attention mechanism, and the constraint for a right stochastic matrix. We report the classification accuracy of our BIMP model with an ablated NOD module on the Cora dataset with 1, 2, 4, 8, 16, 32, 64 and 128 timesteps in Table 14.

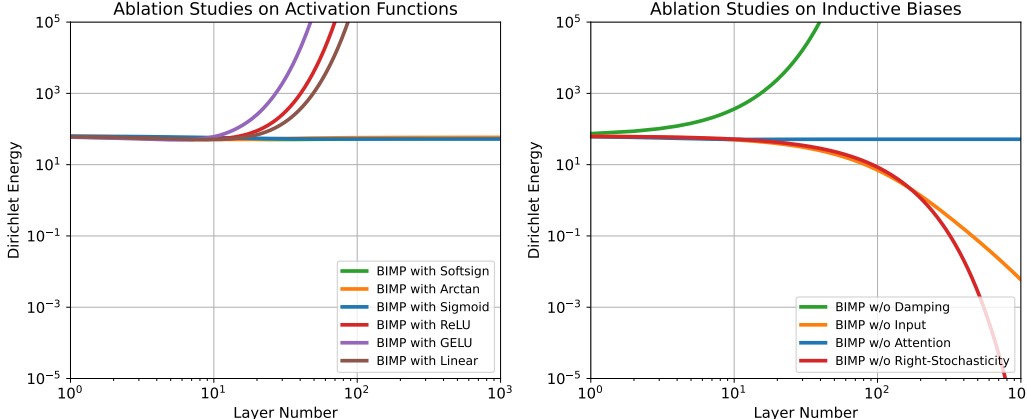

Figure 8: **Dirichlet energy in Ablation study.** Dirichlet Energy evolution of our BIMP model on the synthetic undirected graph. The Dirichlet energy of BIMPs remains stable if they satisfy our theoretical analysis, while the energy of others does not.

BIMP without the input term exhibits oversmoothing, which matches the analysis from Lemma 5.5 and Figure 2 (c), where the weighted sum of the opinion converges to 0 under no input.

BIMP without the damping term also exhibits oversmoothing. Without the damping term our formulation does not satisfy the condition of compact closure (Appendix A.2.4) required to prove Lemma 5.5.

BIMP without an attention mechanism but maintaining the right stochastic adjacency matrix yields results that do not suffer from oversmoothing, since the right stochastic property means Lemma 5.4 still holds. The model however loses expressive capacity and therefore lower classification performance.

BIMP without the right stochastic constraint on the effective adjacency matrix also exhibits oversmoothing. Without the right stochastic constraint, our formulation does not respect the assumptions of Lemma 5.4, and we lose the ability to control the position of the attention parameter and therefore lose the guarantee for oversmoothing characteristics.

For better understanding, we also visualize the Dirichlet energy of node features over a range of network depths, ablating the nonlinear inductive bias. The result is shown in Figure 8 (right). We observe the BIMPs conflicting our proposed lemmas make the Dirichlet energy diverge with depth, while the BIMPs align with the lemmas stabilize the Dirichlet energy.

To summarize, given the two ablation studies in Appendix E.3 and Appendix E.4, we identify the consistent among Dirichlet energy, classification accuracy and our theoretical analysis.

### E.5 PARAMETER SENSITIVITY ANALYSIS

Since the bifurcation-controlled parameter is determined as $u = \frac{d}{\alpha+3}$, governs the emergence of dissensus behavior, ensuring its robustness across different tasks is crucial.

We vary both the damping $d$ and the self-reinforcement $\alpha$ over the range $[0, 5]$ with a step size of 0.5. For each $(d, \alpha)$ pair, we measure the resulting classification accuracy while keeping all other hyperparameters fixed at their fine-tuned values. Each configuration is evaluated over 10 random seeds to ensure statistical reliability. The result is shown in Figure 9.

Recall that our opinion dynamics system is given by

$$\dot{\mathbf{X}} = -d\mathbf{X} + \tanh\left(\frac{d}{\alpha+3}\big((\alpha-1)\mathbf{X} + \tilde{\mathbf{A}}\mathbf{X}\big)\right) + \mathbf{B}. \qquad (118)$$

The classification accuracy deteriorates dramatically when the damping term is set to 0, corresponding to a pure encoder-decoder model where intermediate opinion exchange disappears.

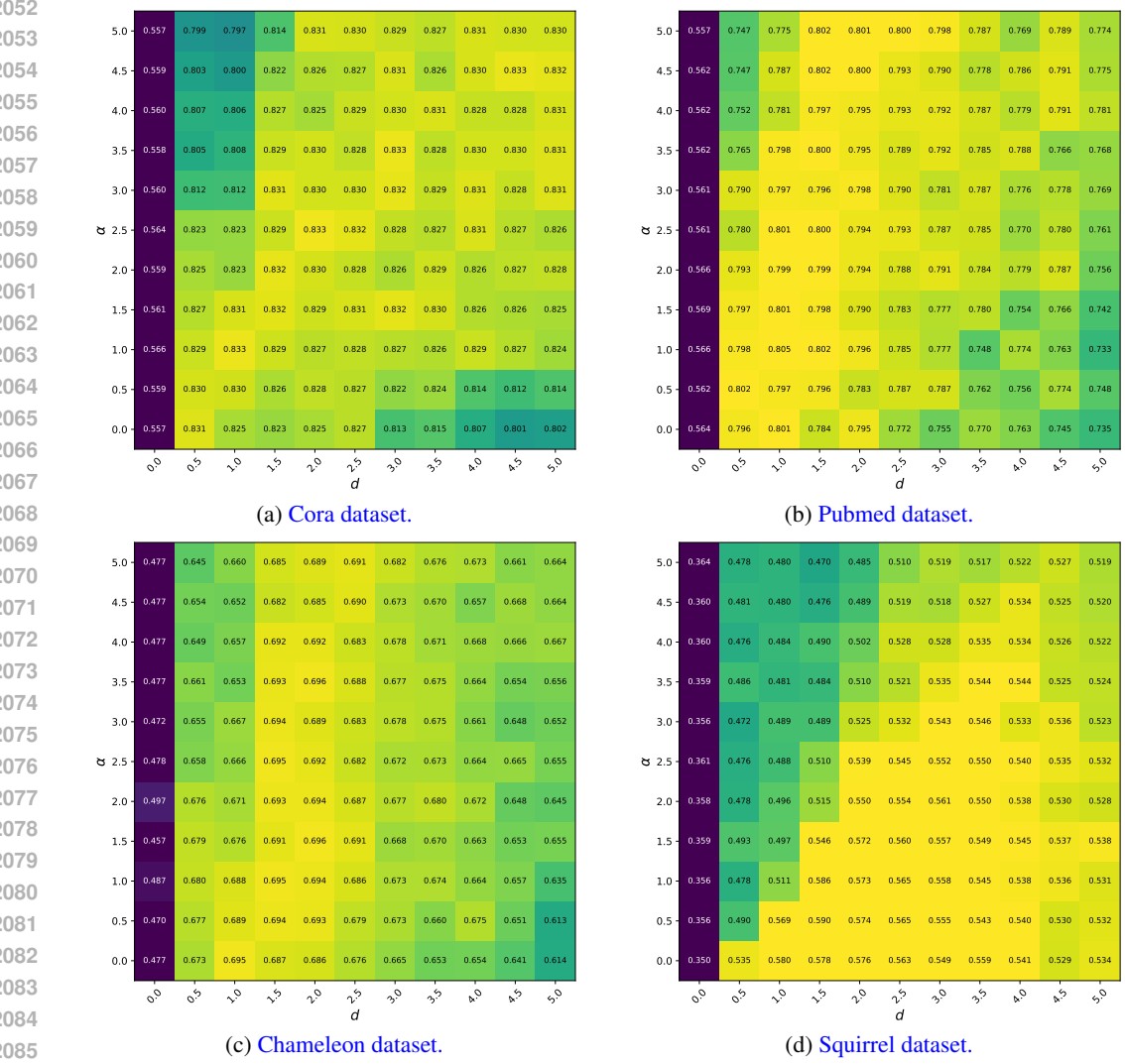

(a) Cora dataset.        (b) Pubmed dataset.

(c) Chameleon dataset.        (d) Squirrel dataset.

Figure 9: **Sensitivity analysis.** The sensitivity map for the attention terms $u = \frac{d}{\alpha+3}$ is shown for both homophilic (Cora and Pubmed) and heterophilic (Chameleon and Squirrel) datasets. The classification accuracy deteriorates dramatically when the damping term is set to 0, which corresponds to a encoder-decoder model where intermediate opinion exchange disappears. The classification accuracy drops slightly at the edges when combinations of $d$ and $\alpha$ reduces the effct of the nonlinear term.

We also observe a slight drop in classification accuracy near the upper-left and lower-right regions of the parameter grid. At the upper-left edge, the attention factor $\frac{d}{\alpha+3}$ is extremely small, leading the nonlinear term $\tanh(\cdot)$ becomes negligible, and the system effectively collapses to a linear system with limited expressive capacity. A similar phenomenon occurs at the lower-right edge, where the large damping $d$ dominates the dynamics. In both cases, the magnitude of the nonlinear component is significantly reduced, leading to a slight degradation in accuracy.

Beyond the highlighted regions, there also exists a broad range of damping $d$ and self-reinforcement $\alpha$ values yield performance that consistently surpasses baselines.

