# OpenReview forum: "Resolving Oversmoothing with Opinion Dissensus"
_ICLR.cc/2026/Conference — Submitted to ICLR 2026_

### Official Review · Reviewer_AqGA · 2025-10-16

**Soundness:** 4
**Presentation:** 4
**Contribution:** 4
**Rating:** 8
**Confidence:** 4

**Summary:**

This paper proposes BIMP (Behavior-Inspired Message Passing), a new class of graph neural networks inspired by nonlinear opinion dynamics. The authors establish a formal analogy between GNN message passing and multi-agent opinion consensus, theoretically showing that oversmoothing in GNNs corresponds to opinion consensus in social dynamics. Based on this insight, they design a nonlinear dynamical system that maintains diversity among node representations. The proposed BIMP model integrates communication and option graphs, introduces nonlinear saturation and bifurcation-controlled parameters, and achieves strong robustness against oversmoothing while remaining computationally efficient.

**Strengths:**

1. Theoretical originality:

    The paper establishes a clear and elegant analogy between oversmoothing in GNNs and opinion consensus in nonlinear dynamics, offering a novel theoretical lens for understanding message passing.

2. Well-motivated nonlinear model:

    The introduction of nonlinear saturation functions, bifurcation-controlled attention, and external inputs provides a principled way to prevent consensus, moving beyond heuristic anti-oversmoothing tricks.

3. Comprehensive analysis and proofs:

    Theoretical lemmas and theorems rigorously support the model’s stability, convergence, and dissensus properties.

4. Strong empirical validation:

    BIMP demonstrates stable Dirichlet energy over 1000 timesteps and consistent superiority across both homophilic and heterophilic datasets, outperforming recent continuous-depth GNNs such as GRAND, GraphCON-Tran, and KuramotoGNN.

**Weaknesses:**

1. Ablation clarity

   The authors should include an additional Dirichlet energy ablation experiment (on activation functions and inductive bias) to verify the impact of different functions and modules on oversmoothing.

Minor issues
* Please provide citations for the baseline models in the main text, and ideally include brief descriptions of these baselines.

**Questions:**

1. To what extent is the robustness to oversmoothing due to the nonlinear opinion dynamics itself, versus architectural choices (e.g., residual terms, attention normalization)?

2. Does the bifurcation-controlled parameter ( $u = \frac{d}{\alpha + 3}$ ) require careful tuning for different datasets, or is it generally robust across tasks?

---

> ### Author Response · Authors · 2025-11-22
>
> > **Include an additional Dirichlet energy ablation experiment to verify the impact of different functions and modules on oversmoothing.**
>
> We have added Dirichlet energy measurements for our choice of nonlinearity (see Appendix E.3) and inductive bias (see Appendix E.4) ablation studies. Appendix E.3, Figure 8 (left) shows that if the nonlinearity in our nonlinear aggregation function does not satisfy the following: $S(0)=0, S'(0)=1, S''(0)\neq0$ [1], the Dirichlet energy of network features increases significantly with depth. Appendix E.4 Figure 8 (right) shows that when using an appropriate nonlinearity, removing any component of the nonlinear opinion dynamics inductive bias causes the Dirichlet energy to diverge with depth. These results are consistent with the trends shown in our accuracy based ablation analysis (Appendix E.3, Table 13 and Appendix E.4, Table 14).
>
> [1] Leonard, Naomi Ehrich, et. al. Annu. rev. control robot. auton. syst. 2024.
>
> > **Is the robustness to oversmoothing due to nonlinear opinion dynamics rather than residual terms, attention or normalization?**
>
> Yes, our ablation study (see Appendix E.3, Table 13) shows the nonlinear inductive bias is necessary even when residual terms (from the input $\mathbf{B}$) and normalization (from the attention mechanism from $\tilde{A}$) are used. Specifically, when we replace the tanh saturation function in our model with the identity function (Table 13, column “linear”), the model performance at deeper layers (e.g., 128) is markedly worse than the performance at shallow layers (e.g., 8).
>
> > **Does the bifurcation-controlled parameter $u=\frac{d}{\alpha+3}$ require careful tuning across tasks?**
>
> Hyperparameter selection, i.e., $d$ and $\alpha$, is robust across tasks. To show this, we performed a parameter sensitivity analysis (see Appendix E.5, Figure 9). In this analysis, we vary both the damping $d$ and the self-reinforcement $\alpha$ over the range $[0, 5]$ with a step size of 0.5. For each $(d, \alpha)$ pair, we report the performance of our network on 2 homophilic datasets and 2 heterophilic datasets. All the other hyperparameters are fixed at their fine-tuned values. Each configuration is evaluated over 10 random seeds to ensure statistical reliability. Our results show that for a broad range of damping $d$ and self‐reinforcement $\alpha$ values, the performance of our BIMP model surpasses baselines.
>
> A notable degradation in BIMP performance occurs for extreme hyperparameter values. For example, when the damping term is set to 0 (this corresponds to a pure encoder-decoder model) intermediate opinion exchange disappears and network performance is poor. We also observe a slight drop in classification accuracy near the upper-left and lower-right regions of the parameter grid. At the upper-left edge, the attention factor $\frac{d}{\alpha + 3}$ is extremely small, making the nonlinear term $\tanh(\cdot)$ negligible. In this case, our nonlinear aggregation term reduces to linear aggregation and network features converge with depth. A similar phenomenon occurs at the lower-right edge, where the large damping $d$ dominates the dynamics.
>
> > **Minor issue: please provide citations for the baseline models in the main text.**
>
> Thank you for pointing out this omission. We have added citations for the baseline models in our main text.

---

> > ### Comment · Reviewer_AqGA · 2025-11-27
> >
> > Thank you for the author's response. I will keep my score.

---

### Official Review · Reviewer_XBcV · 2025-10-29

**Soundness:** 3
**Presentation:** 2
**Contribution:** 3
**Rating:** 4
**Confidence:** 3

**Summary:**

The paper addresses the problem of oversmoothing in GNNs. It makes two significant contributions. First, it proposes a general framework to determine if a GNN model suffers from oversmoothing. In particular, this framework proves oversmoothing for linear discrete-depth GNNs and continuous-depth Laplacian GNNs. This is made possible by mapping a continuous depth GNN to opinion dynamics. Based on this result, the second contribution is a GNN model that does not suffer from oversmoothing. Additionally, the paper tests the performance of the proposed GNN on various datasets, showing slightly better accuracy than state-of-the-art models using similar computational power.

**Strengths:**

The paper presents an innovative connection between GNNs and opinion dynamics. The proposed GNN model is not susceptible to oversmoothing and outperforms state-of-the-art models.

**Weaknesses:**

Some relevant related works may be missing from the paper. For example, there is no reference to how this work relates to previous work on oversmoothing, such as *A Note on Over-Smoothing for Graph Neural Networks by Chen Cai and Yusu Wang* or *Residual Connections and Normalization Can Provably Prevent Oversmoothing in GNNs by Michael Scholkemper, Xinyi Wu, Ali Jadbabaie, and Michael T. Schaub*, which appeared in ICLR 2025. Additionally, there is insufficient reference to discrete-time opinion dynamics, of which there are many nonlinear examples that may adapt better to describe discrete-depth GNNs. See, e.g., *Consensus Dynamics: An Overview by Luca Becchetti, Andrea Clementi, and Emanuele Natale (SIGACT News 2020)*. The relation to these kind works is the subject of Question 3 below.
Finally, it is natural to wonder how the external input B in your paper relate to skip connections (see Question 1).

**Questions:**

1. How does external input B in your paper relate to skip connections, which are known in the literature to prevent oversmoothing?
2. Could the fact that GNNs based on Laplacian dynamics suffer from oversmoothing, even with external input, be an effect of the Laplacian dynamic rather than the linearity of the dynamic?
3. How does your work compare to the references mentioned in the Weaknesses section?

---

> ### Author Response · Authors · 2025-11-22
>
> > **How does external input $\mathbf{B}$ relate to skip connections?**
>
> Our external input $\mathbf{B}$ can be understood as a residual connection and, therefore, a special case of skip connection. While conventional skip connections can mitigate oversmoothing in shallow architectures, our external input $\mathbf{B}$ provably enforces opinion dissensus (see Theorem 5.6) even in very deep architectures (see our empirical results in Appendix D.1.1, Figure 5 and Appendix D.1.2, Figure 6, where we use the skip-connections in ‘-group’).
>
> > **Could the fact that Laplacian-based GNNs suffer from oversmoothing stem from the Laplacian dynamics themselves rather than from the linearity?**
>
> Yes, your intuition about oversmoothing due to Laplacian dynamics is correct! We show that continuous-depth GNNs with Laplacian dynamics (and even for some cases of nonlinear Laplacian dynamics) will exhibit oversmoothing (see Section 4.2 Lemma 4.2). In the case where the Laplacian is constrained to be linear, we show that oversmoothing will occur even in the presence of external inputs (see Section 4.2 Lemma 4.3).
>
> > **How does this work compare to the references mentioned in the Weaknesses section?**
>
> [Chen and Wang, 2020] provide bounds on how fast the Dirichlet energy of GCN features decay with depth. They explore the contribution of the augmented normalized Laplacian, the network's weights, and the network activation functions to the Dirichlet energy. Their core finding is that each of these components can be modified to scale the rate of decay, but none can resolve it. This finding is consistent with the result of Lemma 4.2 in our paper, where we show networks with Laplacian dynamics will exhibit oversmoothing.
>
> [Scholkemper and Wu, et al, 2025] introduce a definition of oversmoothing for which the definitions introduced in [1] and [Chen and Wang, 2020] are special cases. They also prove that, with high probability, residual connections on a properly initialized linear GNN can keep the Dirichlet energy from converging to zero. This finding is complementary to Lemma 4.1 in our paper, where we show linear discrete-depth GNNs (without residual connections) will exhibit oversmoothing.
>
> [Becchetti and Clementi, et al, 2020] consider consensus models (i.e., models for which the Dirichlet energy of agent states tends to zero). In our network, we aggregate node features using a nonlinear opinion dynamics model. This model is more expressive than consensus models and can capture both consensus and dissensus by a change in the bifurcation parameter (see Figure 2).
>
> Thank you for pointing out these related works! We have expanded our discussion to include them.
>
> [1] Oono, Kenta, et al. ICLR 2020.

---

> > ### Comment · Reviewer_XBcV · 2025-11-27
> >
> > I thank the authors for their rebuttal, which clarifies my issues. I raise my score.
> >
> > I would like to point out that I am slightly disappointed with how they added the references I provided to their updated version of the manuscript: they added them here and there, but didn't really add in the manuscript any hint at how their work relate to them, as they do in their rebuttal. This, however, does not affect my score.

---

> > > ### Author Response · Authors · 2025-11-29
> > >
> > > We initially added the references only within the related work section, without explicitly discussing how they connect to our method. In our latest revision, we integrate each reference into our theoretical analysis, specifically:
> > > * [Becchetti and Clementi, 2020] is discussed in Section 3.2: *Linear opinion dynamics*, to highlight the upper bound of the consensus time of linear opinion dynamics models; we also highlight the expressivity of BIMP by comparing it to surveyed consensus models in Section 3.2: *Nonlinear opinion dynamics.*
> > > * [Scholkemper and Wu, 2025] is discussed in Section 4.1 to provide a complementary perspective to Lemma 4.1 in discussing oversmoothing in linearized GNNs.
> > > * [Cai and Wang, 2020] is discussed in Section 4.2 as their bound on rate of decay in Laplacian dynamics aligns with Lemma 4.2.

---

### Official Review · Reviewer_hd9T · 2025-11-01

**Soundness:** 2
**Presentation:** 3
**Contribution:** 2
**Rating:** 2
**Confidence:** 4

**Summary:**

The paper studies the oversmoothing problem in graph neural networks (GNNs) by drawing an analogy to consensus formation in multi-agent opinion dynamics. The authors show that several classes of existing GNNs can be interpreted as equivalent to linear opinion dynamics, which necessarily converge to consensus, and therefore inevitably oversmooth. Based on nonlinear opinion dynamics models, the paper proposes a continuous-depth GNN architecture named BIMP (Behavior-Inspired Message Passing), which is theoretically guaranteed to avoid oversmoothing under certain conditions. Experiments are conducted on common node-classification benchmarks to demonstrate resistance to feature collapse at large depths and competitive accuracy against baselines.

**Strengths:**

1. The analogy between oversmoothing and consensus dynamics is presented clearly and supported by formal statements, which may help unify prior perspectives on oversmoothing.
2. The paper states assumptions and propositions explicitly, with proofs delegated to the appendix, making the theoretical section readable.
3. The empirical plots demonstrate that the proposed model maintains stable Dirichlet energy and accuracy even when simulated for up to 10^3 layers, which supports the main claim of the proposed framework on the oversmoothing side.
4. The paper is easy to follow, and figures illustrating the bifurcation behavior are helpful in conveying the underlying intuition.

**Weaknesses:**

1. Necessity of focusing on oversmoothing is not fully justified.

The paper treats oversmoothing as a central obstacle in GNN design, but it is unclear why controlling depth-induced feature collapse is still an impactful problem in practice. Many modern architectures (e.g., graph transformers, attention-based models with residual paths, or shallow-but-expressive methods) already achieve strong performance with 2–4 layers, often outperforming deeper continuous-depth models regardless of oversmoothing behavior. If a 3-layer model can surpass a 128-step ODE-based model, the motivation for pursuing “arbitrarily deep but stable” GNNs needs stronger justification. The paper would benefit from either (a) a concrete real-world setting where depth improves performance and oversmoothing becomes the bottleneck, or (b) evidence that existing expressive architectures still fail due to oversmoothing rather than other factors (e.g., over-squashing, limited expressivity, memory constraints).

2. Benchmark coverage and presentation choices raise questions.

While the standard homophilic datasets are reported in the main text, several heterophilic results appear only in the appendix, despite being directly relevant to the claim that the model adapts to both regimes via tunable filtering. In addition, the benchmark suite does not explore domains where deep message passing is naturally required (e.g., long-range molecular graphs, multi-hop reasoning, or large-scale relational systems). Since the paper does not claim improvements in runtime, scalability, or memory, the restriction to small- to medium-scale citation graphs feels limiting.

3. Depth stability is demonstrated, but the practical benefit is unclear.

The paper shows that BIMP avoids Dirichlet-energy collapse and remains trainable at large depths, but it is not shown why this matters for downstream tasks. In the current benchmarks, deeper baselines typically do not fail catastrophically—they just plateau or degrade moderately. An illustrative failure case, where a strong baseline collapses due to oversmoothing and BIMP succeeds, would make the motivation more concrete.

4. Novelty is mainly conceptual rather than architectural.

Although the theoretical analogy is interesting, the empirical model resembles prior continuous-depth GNNs with a particular nonlinear choice and learned adjacency. The extent to which performance gains arise from the opinion-dynamics design rather than standard architectural components is not fully disentangled.

5. Scope limited to oversmoothing, not broader depth-related limitations.

Other known depth challenges—such as over-squashing, expressivity limits, and scaling to large graphs—are not addressed. Since oversmoothing is only one of several depth-related bottlenecks, the contribution feels narrow unless its relevance can be better contextualized.

**Questions:**

1. Can the authors provide a concrete example where oversmoothing is the dominant failure mode in modern GNNs?
2. How does BIMP perform on domains where depth is required (e.g., long-range molecular interaction graphs, knowledge graphs, large-scale recommender systems)?
3. Is the nonlinear inductive bias still necessary when skip-connections, normalization, or positional encodings are added? An ablation isolating the source of benefit would be useful.
4. How does BIMP compare against non-ODE architectures that already mitigate oversmoothing implicitly?
4. Since nonlinear continuous-depth models are not the only way to address the scenarios tested in the paper, how are the runtime / memory comparisons against strong non-ODE baselines (e.g., GCN with residuals, GraphGPS, GATv2, or GNN-SSM), especially on larger graphs.

---

> ### Author Response · Authors · 2025-11-22
> **Response to Reviewer hd9T (1/2)**
>
> > **Can the authors provide a concrete example where oversmoothing is the dominant failure mode in modern GNNs?**
>
> Oversmoothing is the dominant failure mode in GNN based multi-agent trajectory prediction. In this task, a GNN model is recurrently applied to agent features which can be seen as equivalent to stacking many layers (see Appendix E.1.2 for more details). In general, to succeed in long-horizon prediction, the model must be able to produce diverging trajectories at deep depths, which is not possible when there is oversmoothing.
>
> We provide quantitative and qualitative comparisons of our model and the GCN-based GraphODE in Table 10 and Figure 4. In Table 10, we report the MSE and MAPE prediction error of our model and GraphODE for short and long horizon prediction. Our model shows significantly lower prediction error for all cases. In Figure 4 we offer a qualitative comparison of the predicted trajectories and learned features of our model and GraphODE. On the left we see the larger MSE error of GraphODE results from the convergence of agent trajectories. On the right we see the convergence of agent trajectories results from the feature consensus (i.e., the learned trajectories become increasingly similar in long horizon prediction). The visual trend is quantified by measuring the variance of each feature dimension across agents.
>
> > **How does BIMP perform on domains where depth is required (e.g., long-range molecular interaction graphs, knowledge graphs, large-scale recommender systems)**
>
> To demonstrate the performance of our model when depth is required, we apply our BIMP model to long-range molecular prediction. Specifically, we selected two tasks from the Long Range Graph Benchmark (LRGB) [1]: Peptides-func and Peptides-struct, corresponding to graph classification and regression, respectively. This dataset is well-suited for assessing long-range reasoning, as its graphs exhibit notably large shortest-path distances (≈20) and diameters (≈57), requiring models to effectively exchange information across distant nodes. We report the result here and in our paper (Table. 3). Due to time constraints, baseline results were taken directly from [2].
>
> | Model | Peptides-func (AP ↑) | Peptides-struct (MAE ↓) |
> |-|-|-|
> | GCN | 59.30 ± 0.23 | 0.3496 ± 0.0013 |
> | GIN [3] | 54.98 ± 0.79 | 0.3547 ± 0.0045 |
> | Transformer + LapPE [4] | 63.26 ± 1.26 | **0.2529 ± 0.0016** |
> | SAN + LapPE [5] | **63.84 ± 1.21** | 0.2683 ± 0.0043 |
> | GRAND-$\ell$ | 57.89 ± 0.62 | 0.3418 ± 0.0015 |
> | GraphCON | 60.22 ± 0.68 | 0.2778 ± 0.0018 |
> | ADGN [6] | 59.75 ± 0.44 | 0.2874 ± 0.0021 |
> | **BIMP (ours)** | **63.62 ± 1.07** | **0.2629 ± 0.0027** |
>
> Our BIMP model outperforms all the continuous-depth models and performs on par with Graphormers without requiring time consuming preprocessing (on the Peptides-struct dataset, preprocessing for LapPE takes 1 min 14 s on a single NVIDIA A100 with four AMD Milan 7413 CPU cores [1]). Moreover, BIMP is more parameter efficient, requiring only $\sim$111K parameters, compared to the $\sim$500K parameters required by Graphormers (i.e., Transformer+LapPE and SAN+LapPE).
>
> [1] Dwivedi, Vijay Prakash, et al. Neurips 2022.
>
> [2] Gravina, Alessio, et al. AAAI 2025.
>
> [3] Xu, Keyulu, et al. ICLR 2019.
>
> [4] Dwivedi, Vijay Prakash, et al. AAAI-Workshop 2020.
>
> [5] Kreuzer, Devin, et al. NeuIPS 2021.
>
> [6] Gravina, Alessio, et al. ICLR 2023.
>
> > **Is the nonlinear inductive bias still necessary when skip-connections, normalization, or positional encodings are added? An ablation isolating the source of benefit would be useful.**
>
> Yes, our ablation study (see Appendix E.3, Table 13) shows the nonlinear inductive bias is necessary even when skip-connections (from the input $\mathbf{B}$) or normalization (from the attention mechanism from $\tilde{A}$) are used. Specifically, when we replace the tanh saturation function in our model with the identity function (Table 13, column “linear”), the model performance at deeper layers (e.g., 128) is markedly worse than the performance at shallow layers (e.g., 8).
>
> > **How does BIMP compare against strong non-ODE architectures that already mitigate oversmoothing implicitly?**
>
> We compare BIMP to non-ODE architectures with Pairnorm and differentiable group normalization (DGN) in Appendix D.1.1, Figure 5 and 6, and Appendix D.2.1, Table 4. Our results show that Pairnorm and DGN only mitigate oversmoothing (and performance drop) at shallow depths. We have updated the text in the main body to clarify that the experiments in Appendix D.1.1, Figure 5 and 6, and Appendix D.2.1, Table 4 are against non-ODE baselines that use oversmoothing mitigation techniques.

---

> ### Author Response · Authors · 2025-11-22
> **Response to Reviewer hd9T (2/2)**
>
> > **Since nonlinear continuous-depth models are not the only way to address the scenarios tested in the paper, how are the runtime / memory comparisons against strong non-ODE baselines (e.g., GCN with residuals, GraphGPS, GATv2, or GNN-SSM), especially on larger graphs.**
>
> We have added runtime and memory usage comparisons with non-ODE baselines (i.e., GCN with residuals[1], GATv2[2] and GOAT[3]) on the large graph dataset ogbn-arXiv. We compare against GOAT (a Graph Transformer with positional encodings that mitigates oversmoothing through multi-hop feature propagation) instead of GraphGPS[4] since GraphGPS does not provide an implementation for node classification. We also do not include GNN-SSM[5] as the official code has not yet been released. The results of our comparisons are reported here and in Appendix E.2, Table 12 of our paper. Please note the numbers in parentheses indicate the additional time for positional encoding preprocessing.
>
> || BIMP(ours) | GCN-residual | GATv2 | GOAT |
> |-|-|-|-|-|
> | Running time (s) | 114.15 | 37.43 | 77.81 | 3601.13 (75.96) |
> | Memory usage (MB) | 1465 | 4628 | 6975 | 3210 |
>
> Our runtime is slightly higher than GCN with residual connections and GATv2, but our memory usage is lower than all baseline architectures. This suggests that most of the computational cost for our method comes from the ODE solver.
>
> Additionally, we compared the runtime of BIMP against ODE baselines in Appendix E.2, Table 11. In contrast to other nonlinear methods like KuramotoGNN, the training time of BIMP is comparable to other linear ODE baselines.
>
> [1] Chi, Huixuan, et al. Journal of Physics: Conference Series 2022.
>
> [2] Brody, Shaked, et al. ICLR 2022.
>
> [3] Kong, Kezhi, et al. ICML 2023.
>
> [4] Rampášek, Ladislav, et al. Neurips 2022.
>
> [5] Arroyo, Álvaro, et al. Neurips 2025.
>
> > **Extend the scope to other depth-related limitations, including over-squashing, expressivity limits, and scaling to large graphs.**
>
> We discuss expressivity constraints in what is now Theorem 5.8, scalability to large graphs in Appendix D.2.3, Table 7, and broaden the scope of our paper on over-squashing in Section 6.2.
>
> Regarding expressivity, our model is provably more expressive than linear opinion dynamics models. In the proof of Theorem 5.8 (previously Appendix B, Lemma B.1) we show that linear opinion dynamics is a first-order approximation of BIMP. Moreover, unlike many continuous-depth GNNs (e.g., GRAND, GRAND++) which lack a feature mixing mechanism, BIMP enables mixing across feature dimensions enabling more complex information exchange.
>
> Regarding scalability to large graphs, our model achieves superior classification performance compared to baseline methods while using a relatively small number of parameters. We report the classification performances on the ogbn-arXiv in Appendix D.2.3, Table 7, which is a large graph with 169,343 nodes and 1,166,243 edges.
>
> We have broadened the scope of our paper to include a discussion on over-squashing (see Section 6.2) and include new empirical results to demonstrate the effectiveness of our approach on long range prediction tasks  (see our response to Question 2). We compare BIMP to baseline models on the Long Range Graph Benchmark[1]. BIMP achieves strong performance on two long-range molecular prediction tasks, indicating its effectiveness in mitigating over-squashing.
>
> [1] Dwivedi, Vijay Prakash, et al. Neurips 2022.

---

### Author Response · Authors · 2025-11-22

We thank the reviewers for their time and constructive feedback. We are encouraged that the reviewers found our proposed BIMP model to be innovative, well motivated, and clearly presented (hd9T, XBcV, AqGA); supported by formal proofs and analysis (hd9T, AqGA); resistant to oversmoothing at large depths (hd9T, XBcV); and performant against baselines methods (AqGA, XBcV).

In response to reviewer questions about the source of robustness to oversmoothing (hd9T, XBcV, AqGA), we highlight our experimental results in Figure 5, Figure 6 and Table 13. Comparisons of classification accuracy (Figure 5) and Dirichlet energy (Figure 6) demonstrate that architectural choices alone (i.e., normalization and skip-connections in Pairnorm and DGN), cannot mitigate oversmoothing at large depth; and our ablation study (Table 13) shows that BIMP becomes susceptible to oversmoothing when the nonlinearity in our aggregation function is removed (even if residuals and attention are retained). This strongly suggests the robustness of our architecture is due to our inductive bias and not conventional architectural choices.

In response to reviewer hd9T’s request for a concrete real-world experiment where oversmoothing becomes the bottleneck, we highlight our multi-agent trajectory prediction task. We added a new visualization of latent features and their variance in Figure 4, and observe that for baseline models, the latent representations become similar for all agents resulting in similar next state predictions. These results strengthen our motivation for resolving oversmoothing, and highlight our model’s superior performance.

In response to reviewer hd9T’s question about resilience to oversquashing, we have added new results on the long-range molecular graph benchmark (Peptides-func and Peptides-struct datasets). In Table 3 we show that our model performs on par with state-of-the-art models, while requiring fewer parameters, shorter training time, and lower memory usage. Thank you for pointing out this opportunity to broaden the scope of our paper to over-squashing, which is a fundamental bottleneck that restricts long-range information.

In response to reviewer AqGA’s question about the robustness of our model to hyperparameter selection, we have added a new sensitivity analysis. In Appendix E.5, Figure 9, we include results on 2 homophilic datasets (Cora, Pubmed) and 2 heterophilic datasets (Chameleon and Squirrel). Thank you for suggesting this opportunity to show that our model is performant for a broad range of damping $d$ and self‐reinforcement $\alpha$ values.

We address other reviewer comments below and have incorporated all feedback in our rebuttal revision. We highlight changes to our original manuscript with blue text.

---

### Author Response · Authors · 2025-12-03
**Summary for the new area chair**

Dear AC,

We appreciate your time and effort in dealing with this unprecedented situation. To facilitate your final decision, we provide a concise summary of the rebuttal process.

**Initial Reviews**: all the initial reviews were constructive. Two reviewers (XBcV, AqGA) highlighted our novelty and strong performance against baselines, but suggested further clarifications. One reviewer (hd9T) acknowledged our method's resistance to oversmoothing, but requested additional experiments to reinforce the motivation and scope of our work.

**Addressing concerns with clarifications**: we clarify our method to address the following points.

* Source of robustness to oversmoothing (hd9T, XBcV, AqGA). We highlight our experimental results in Table 13, where ablating our nonlinearity results in oversmoothing even in the presence of mitigation techniques such as residual connection and attention mechanism.

* Expressivity and scalability (hd9T). We highlight Theorem 5.8 (previously presented in the appendix), showing our method has greater expressivity compared to baselines such as GRAND. We also demonstrate scalability with strong performance on the large dataset ogbn-arxiv (Table 7).

* Robustness of hyperparameter selection (AqGA). We performed a parameter sensitivity analysis (Figure 9), showing consistent performance over a broad range of hyperparameter values.

* Minor clarification on related works and visualizations (XBcV, AqGA). We incorporated all suggested references in our revision. Additionally, we provided a new Dirichlet energy visualization (Figure 8) in our ablation results.

These clarifications are well-received: **the active reviewers stated that either their concerns were resolved and updated their score accordingly, or maintained a positive view on the paper (details are in their official response).**

**Reinforcing the motivation and scope with additional experiments**: to resolve the main concerns of reviewer hd9T, we conduct 3 additional experiments.

* Trajectory prediction task with analysis of latent feature variance. In response to the request for a concrete real-world experiment where oversmoothing becomes the bottleneck, we highlight our multi-agent trajectory prediction task (Figure 1 and Appendix E.1). We add a new visualization of latent features and their variance in Figure 4, where we quantified oversmoothing behavior in latent features. Our method achieves superior performance by effectively overcoming oversmoothing.

* Long-range graph tasks. To address concerns about tasks where increased depth is required, we introduced new evaluations on long-range molecular graph benchmarks (Peptides-func and Peptides-struct), where BIMP achieves SOTA performance. The results demonstrate our method’s adaptability to long range as well as its flexibility to address oversquashing.

* Runtime and memory comparison. To address concerns about computational cost, we compare the runtime and memory usage of our method against strong non-ODE baselines, including GCN with residual connections, GATv2, and GOAT. The results (Table 12) show that our approach achieves comparable runtime while maintaining strong memory efficiency.

In conclusion, **we believe that we comprehensively addressed every question raised by the reviewers**, refined our manuscript to include the key additional results requested, and got strong support from the reviewers who engaged with the discussion.

We hope this summary can help you make the final decision.

---

### Meta-Review · Area_Chair_SuYy · 2026-01-16

**Summary:**

The paper aims to propose a way to get rid of well known over-smoothing issue in GNN by connecting and borrowing tools from opinion dynamics. It is a borderline paper. Reviewers asked for several clarifications including comparing with related work and bench-marking with  graph transformers and other architecture. The results show some of the baselines are comparable to the proposed method. So it is unclear what benefit the proposed method is providing with.  Upon carefully re-looking in the paper, I observed that even in Table 4 and 5, the difference between proposed method and current method marginal. It would be great if the authors provide a statistical significance test to prove the efficacy of their method.

Oversmoothing in GNN is long discussed in literature but this method has some merit. I will recommend that the authors provide a different narrative for the paper, possibly revealing some connection between opinion dynamics and GNN would be very helpful.
I think the paper should provide more comprehensive evaluation and benchmarking in the main paper.

**Reviewer Concerns:**

Authors attempted to address the concerns. However, some of the experimental comparison casts doubt on utility of the underlying method.

**Reviewer Scores:**

Reviewer hd9T may not have increased the score since the performance improvement compared to new baselines is not sufficient.

---

### Decision · Program_Chairs · 2026-01-26

Reject